# THREE FACTORS INFLUENCING MINIMA IN SGD

## ABSTRACT

We study the statistical properties of the endpoint of stochastic gradient descent (SGD). We approximate SGD as a stochastic differential equation (SDE) and consider its Boltzmann Gibbs equilibrium distribution under the assumption of isotropic variance in loss gradients.. Through this analysis, we find that three factors – learning rate, batch size and the variance of the loss gradients – control the trade-off between the depth and width of the minima found by SGD, with wider minima favoured by a higher ratio of learning rate to batch size. In the equilibrium distribution only the ratio of learning rate to batch size appears, implying that it's invariant under a simultaneous rescaling of each by the same amount. We experimentally show how learning rate and batch size affect SGD from two perspectives: the endpoint of SGD and the dynamics that lead up to it. For the endpoint, the experiments suggest the endpoint of SGD is similar under simultaneous rescaling of batch size and learning rate, and also that a higher ratio leads to flatter minima, both findings are consistent with our theoretical analysis. We note experimentally that the dynamics also seem to be similar under the same rescaling of learning rate and batch size, which we explore showing that one can exchange batch size and learning rate in a cyclical learning rate schedule. Next, we illustrate how noise affects memorization, showing that high noise levels lead to better generalization. Finally, we find experimentally that the similarity under simultaneous rescaling of learning rate and batch size breaks down if the learning rate gets too large or the batch size gets too small.

## 1 INTRODUCTION

Despite being massively over-parameterized (Zhang et al., 2016), deep neural networks (DNNs) have demonstrated good generalization ability and achieved state-of-the-art performances in many application domains such as image (He et al., 2016) and speech recognition (Amodei et al., 2016). The reason for this success has been a focus of research recently but still remains an open question. Our work provides new theoretical insights and useful suggestions for deep learning practitioners.

The standard way of training DNNs involves minimizing a loss function using SGD and its variants (Bottou, 1998). In SGD, parameters are updated by taking a small discrete step depending on the learning rate in the direction of the negative loss gradient, which is approximated based on a small subset of training examples (called a mini-batch). Since the loss functions of DNNs are highly non-convex functions of the parameters, with complex structure and potentially multiple minima and saddle points, SGD generally converges to different regions of parameter space depending on optimization hyper-parameters and initialization.

Recently, several works (Arpit et al., 2017; Advani & Saxe, 2017; Shirish Keskar et al., 2016) have investigated how SGD impacts generalization in DNNs. It has been argued that wide minima tend to generalize better than sharp minima (Hochreiter & Schmidhuber, 1997; Shirish Keskar et al., 2016). This is entirely compatible with a Bayesian viewpoint that emphasizes targeting the probability mass associated with a solution, rather than the density value at a solution (MacKay, 1992b). Specifically, (Shirish Keskar et al., 2016) find that larger batch sizes correlate with sharper minima. In contrast, we find that it is the ratio of learning rate to batch size which is correlated with sharpness of minima, not just batch size alone. In this vein, while (Dinh et al., 2017) discuss the existence of sharp minima which behave similarly in terms of predictions compared with wide minima, we argue that SGD naturally tends to find wider minima at higher noise levels in gradients, and such wider minima seem to correlate with better generalization.

In order to achieve our goal, we approximate SGD as a continuous stochastic differential equation (Bottou, 1991; Mandt et al., 2017; Li et al., 2017). Assuming isotropic gradient noise, we derive the Boltzmann-Gibbs equilibrium distribution of this stochastic process, and further derive the relative probability of landing in one local minima as compared to another in terms of their depth and width. Our main finding is that the ratio of learning rate to batch-size along with the gradient's covariances influence the trade-off between the depth and sharpness of the final minima found by SGD, with a high ratio of learning rate to batch size favouring flatter minima. In addition, our analysis provides a theoretical justification for the empirical observation that scaling the learning rate linearly with batch size (up to a limit) leads to identical performance in DNNs (Krizhevsky, 2014; Goyal et al., 2017).

We verify our theoretical insights experimentally on different models and datasets. In particular, we demonstrate that high learning rate to batch size ratio (due to either high learning rate or low batch-size) leads to wider minima and correlates well with better validation performance. We also show that a high learning rate to batch size ratio helps prevent memorization. Furthermore, we observe that multiplying each of the learning rate and the batch size by the same scaling factor results in similar training dynamics. Extending this observation, we validate experimentally that one can exchange learning rate and batch size for the recently proposed cyclic learning rate (CLR) schedule (Smith, 2015), where the learning rate oscillates between two levels. Finally, we discuss the limitations of our theory in practice.

## 2   RELATED WORK

The relationship between SGD and sampling a posterior distribution via stochastic Langevin methods has been the subject of discussion in a number of papers (Chaudhari et al., 2017; Chen et al., 2014; Ding et al., 2014; Vollmer et al., 2015; Welling & Teh, 2011; Shang et al., 2015; Sato & Nakagawa, 2014). In particular, (Mandt et al., 2017) describe the dynamics of stochastic gradient descent (SGD) as a stochastic process that can be divided into three distinct phases. In the first phase, weights diffuse and move away from the initialization. In the second phase the gradient magnitude dominates the noise in the gradient estimate. In the final phase, the weights are near the optimum. (Shwartz-Ziv & Tishby, 2017) make related observations from an information theoretic point of view and suggest the diffusion behaviour of the parameters in the last phase leads to the minimization of mutual information between the input and hidden representation. We also relate the SGD dynamics to the stationary distribution of the stochastic differential equation. Our derivation bears similarity with (Mandt et al., 2017). However, while (Mandt et al., 2017) study SGD as an approximate Bayesian inference method in the final phase of optimization in a locally convex setting, our end goal is to analyze the stationary distribution over the entire parameter space reached by SGD. Further, our analysis allows us to compare the probability of SGD ending up in one minima over another (in terms of width and depth), which is novel in our case.

We discuss the Fokker-Planck equation which has appeared before in the machine learning literature though the exact form and solution we consider we believe is novel. For example, in the online setting (Heskes & Kappen, 1993) derive a Gibbs distribution from the Fokker-Planck equation, but the relation there does not give the temperature of the Gibbs distribution in terms of the learning rate, batch size and gradient covariance.

Our work is also closely related to the ongoing discussion about the role of large batch size and the sharpness of minima found in terms of generalization (Shirish Keskar et al., 2016). (Shirish Keskar et al., 2016) showed that SGD ends up in sharp minimum when using large batch size. (Goyal et al., 2017; Hoffer et al., 2017) empirically observed that scaling up the learning rate, and training for more epochs, leads to good generalization when using large batch size. Our novelty is in explaining the importance of the ratio of learning rate to batch size. In particular, our theoretical and empirical results show that simultaneously rescaling the batch size and learning rate by the same amount leads SGD to minima having similar width despite using different batch sizes.

Concurrent with this work, (Smith & Le, 2017; Chaudhari & Soatto, 2017) have both analyzed SGD approximated as a continuous time stochastic process and stressed the importance of the learning rate to batch size ratio. (Smith & Le, 2017) focused on the training dynamics while (Chaudhari & Soatto, 2017) explored the stationary non-equilibrium solution for the stochastic differential equation for non-isotropic gradient noise, but assuming other conditions on the covariance and loss to

enforce the stationary distribution to be path-independent. Their solution does not have an explicit solution in terms of the loss in this case. In contrast to other work, we strictly focus on the explicitly solvable case of the Boltzmann-Gibbs equilibrium distribution with isotropic noise. This focus allows us to relate the noise in SGD, controlled by the learning rate to batch size ratio, with the width of its endpoint. We empirically verify that the width and height of minima correlates with the learning rate to batch size ratio in practice.

Our work continues the line of research on the importance of noise in SGD (Bottou, 1998; Roux et al., 2008; Neelakantan et al., 2015; Mandt et al., 2017). Our novelty is in formalizing the impact of batch size and learning rate (i.e. noise level) on the width and depth of the final minima, and empirical verifications of this.

## 3 INSIGHTS FROM FOKKER-PLANCK

Our focus in this section is on finding the relative probability with which we end optimization in a region near a minimum characterized by a certain loss value and Hessian determinant. We will find that the relative probability depends on the local geometry of the loss function at each minimum along with batch size, learning rate and the covariance of the loss gradients. To reach this result, we first derive the equilibrium distribution of SGD over the parameter space under a stochastic differential equation treatment. We make the assumption of isotropic covariance of the loss gradients, which allows us to write down an explicit closed-form analytic expression for the equilibrium distribution, which turns out to be a Boltzmann-Gibbs distribution.

### 3.1 SETUP

We follow a similar (though not identical) theoretical setup to (Mandt et al., 2017), approximating SGD with a continuous-time stochastic process, which we now outline.

Let us consider a model parameterized by $\boldsymbol{\theta} = \{\theta_1, \dots, \theta_q\}$. For $N$ training examples $\boldsymbol{x}_i, i \in \{1, ..., N\}$, the loss function, $L(\boldsymbol{\theta})$, and the corresponding gradient $\mathbf{g}(\boldsymbol{\theta})$, are defined based on the sum over the loss values for all training examples. Stochastic gradients $\mathbf{g}^{(S)}(\boldsymbol{\theta})$ arise when we consider a batch $\mathcal{B}$ of size $S < N$ of random indices drawn uniformly from $\{1, ..., N\}$ and form an (unbiased) estimate of loss and gradient based on the corresponding subset of training examples

$$L^{(S)}(\boldsymbol{\theta}) = \frac{1}{S} \sum_{n \in \mathcal{B}} l(\boldsymbol{\theta}, \boldsymbol{x}_n) , \qquad\qquad \mathbf{g}^{(S)}(\boldsymbol{\theta}) = \frac{\partial}{\partial \boldsymbol{\theta}} L^{(S)}(\boldsymbol{\theta}) .$$

We consider stochastic gradient descent (SGD) with learning rate $\eta$, as defined by the update rule

$$\boldsymbol{\theta}(t+1) = \boldsymbol{\theta}(t) - \eta \boldsymbol{g}^{(S)}(\boldsymbol{\theta}) .$$

We now make the following assumptions:

(1) By the central limit theorem (CLT), we assume the noise in the stochastic gradient is Gaussian with covariance matrix $\frac{1}{S}\mathbf{C}(\boldsymbol{\theta})$

$$\mathbf{g}^{(S)}(\boldsymbol{\theta}) = \mathbf{g}(\boldsymbol{\theta}) + \frac{1}{\sqrt{S}}\Delta\mathbf{g}(\boldsymbol{\theta}), \text{ where } \Delta\mathbf{g}(\boldsymbol{\theta}) \sim N(0, \mathbf{C}(\boldsymbol{\theta})) .$$

We note that the covariance is symmetric positive-semidefinite, and so can be decomposed into the product of two matrices $\mathbf{C}(\boldsymbol{\theta}) = \mathbf{B}(\boldsymbol{\theta})\mathbf{B}^\top(\boldsymbol{\theta})$ .

(2) We assume the discrete process of SGD can be approximated by the continuous time limit of the following stochastic differential equation (known as a Langevin equation)

$$\frac{d\boldsymbol{\theta}}{dt} = -\eta g(\boldsymbol{\theta}) + \frac{\eta}{\sqrt{S}}\mathbf{B}(\boldsymbol{\theta})\mathbf{f}(t) \tag{1}$$

where $\mathbf{f}(t)$ is a normalized Gaussian time-dependent stochastic term.

Note that the continuous time approximation of SGD as a stochastic differential equation has been shown to hold in a weak approximation on the condition that the learning rate is small (Li et al., 2017).

Note that we have not made Assumption 4 of (Mandt et al., 2017), where they assume the loss can be globally approximated by a quadratic. Instead, we allow for a general loss function, which can have many local minima.

## 3.2 THREE FACTORS INFLUENCING EQUILIBRIUM DISTRIBUTION

The Langevin equation is a stochastic differential equation, and we are interested in its equilibrium distribution which gives insights into the behavior of SGD and the properties of the points it converges to. Assuming isotropic noise, the Langevin equation is well known to have a Gibbs-Boltzmann distribution as its equilibrium distribution. This equilibrium distribution can be derived by finding the stationary solution of the Fokker-Planck equation, with detailed balance, which governs the evolution of the probability density of the value of the parameters with time. The Fokker-Planck equation and its derivation is standard in the statistical physics literature. In Appendix A we give the equation in the machine learning context in Eq. (5) and for completeness of presentation we also give its derivation. In Appendix C we restate the standard proofs of the stationary distribution of a Langevin system, and provide the resulting Gibbs-Boltzmann equilbirium distribution here, using the notation of this paper:

**Theorem 1** (Equilibrium Distribution). *Assume*[1] *that the gradient covariance is isotropic, i.e.* $\mathbf{C}(\boldsymbol{\theta}) = \sigma^2 \mathbf{I}$, *where* $\sigma^2$ *is a constant. Then the equilibrium distribution of the stochastic differential equation 1 is given by*

$$P(\boldsymbol{\theta}) = P_0 \exp \left( -\frac{2L(\boldsymbol{\theta})}{n\sigma^2} \right) \quad , \tag{2}$$

*where* $n \equiv \frac{\eta}{S}$ *and* $P_0$ *is a normalization constant, which is well defined for loss functions with* $L_2$ *regularization.*

**Discussion**: Here $P(\boldsymbol{\theta})$ defines the density over the parameter space. The above result says that if we run SGD for long enough (under the assumptions made regarding the SGD sufficiently matching the infinitesimal limit), then the probability of the parameters being in a particular state asymptotically follows the above density. Note, that $n \equiv \frac{\eta}{S}$ is a measure of the noise in the system set by the choice of learning rate $\eta$ and batch size $S$. The fact that the loss is divided by $n$ emphasizes that the higher the noise $n$, the less granular the loss surface *appears* to SGD. The gradient variance $\mathbf{C}(\boldsymbol{\theta})$ on the other hand is determined by the dataset and model priors (e.g. architecture, model parameterization, batch normalization etc.). This reveals an important area of investigation, i.e., how different architectures and model parameterizations affect the gradient's covariance structure. We note that in the analysis above, the assumption of the gradient covariance $\mathbf{C}(\boldsymbol{\theta})$ being fixed and isotropic in the parameter space is unrealistic. However it is a simplification that enables straightforward insights regarding the relationship of the noise, batch size and learning rate in the Gibbs-Boltzmann equilibrium. We empirically show that various predictions based on this relationship hold in practice.

Returning to SGD as an optimization method, we can ask, given the probability density $P(\boldsymbol{\theta})$ can we derive the probability of ending at a given minimum, $\boldsymbol{\theta}_A$, which we will denote by lowercase $p_A = \tilde{p}_A C$, where $C$ is a normalization constant which is the same for all minima (the unnormalized probability $\tilde{p}_A$ is all we are interested in when estimating the relative probability of finishing in a given minimum compared to another one). This probability is derived in Appendix D, and given in the following theorem, which is the core insight from our theory.

**Theorem 2** (Probability of ending in region near minima $\boldsymbol{\theta}_A$). *Assume the loss has a series of separated local minima. Consider one such minima, with Hessian* $\mathbf{H}_A$ *and loss* $L_A$ *at a minimum* $\boldsymbol{\theta}_A$. *Then the unnormalized probability of ending in a region near minima* $\boldsymbol{\theta}_A$ *is*

$$\tilde{p}_A = \frac{1}{\sqrt{\det \mathbf{H}_A}} \exp \left( -\frac{2L_A}{n\sigma^2} \right) \tag{3}$$

*where* $n = \frac{\eta}{S}$ *is the noise used in the SGD process to reach* $\theta_A$.

**Discussion**: For this analysis, we qualitatively categorize a minima $\boldsymbol{\theta}_A$ by its loss $L_A$ (depth) and the determinant of the Hessian $\det \mathbf{H}_A$ (a larger determinant implies a sharper minima). The above

---

[1]Here we also assume a weak regularity condition that the loss $L(\boldsymbol{\theta})$ includes the regularization term $\tau \|\boldsymbol{\theta}\|_2^2$ for some $\tau > 0$.

result shows that the probability of landing in a specific minimum depends on three factors - learning rate, batch-size and covariance of the gradients. The two factors that we directly control only appear in the ratio given by the noise $n = \eta/S$. Note that the proof of this result utilizes a Laplace Approximation in which the loss near a given minimum can be approximated using a second order Taylor series in order to evaluate an integral. We emphasize this is not the same as globally treating the loss as a quadratic.

To study which kind of minima are more likely if we were to reach equilibrium, it is instructive to consider the ratio of probabilities $p_A$ and $p_B$ at two distinct minima $\boldsymbol{\theta}_A$ and $\boldsymbol{\theta}_B$ respectively given by

$$\frac{p_A}{p_B} = \sqrt{\frac{\det \mathbf{H}_B}{\det \mathbf{H}_A}} \exp \left( \frac{2}{n\sigma^2} \left( L_B - L_A \right) \right) \quad .$$

To highlight that towards the equilibrium solution SGD favors wider rather than sharper minima, let's consider the special case when $L_A = L_B$, i.e., both minima have the same loss value. Then,

$$\frac{p_A}{p_B} = \sqrt{\frac{\det \mathbf{H}_B}{\det \mathbf{H}_A}}.$$

This case highlights that in equilibrium, SGD favors the minimum with lower determinant of the Hessian (i.e. the flatter minima) when all other factors are identical. On the flip side, it can be seen that if two minima have the same curvature ($\det \mathbf{H}_A = \det \mathbf{H}_B$), then SGD will favor the minima with lower loss. Finally in the general case when $L_A \geq L_B$, it holds that $p_A \geq p_B$ if and only if

$$\frac{1}{n} \leq \frac{2\sigma^2 \log \left( \sqrt{\frac{\det \mathbf{H}_B}{\det \mathbf{H}_A}} \right)}{(L_A - L_B)} \equiv Y \quad .$$

That is, there is an upper bound on the inverse of the noise for $\boldsymbol{\theta}_A$ to be favored in the case that its loss is higher than at $\boldsymbol{\theta}_B$, and this upper bound depends on the difference in the heights compared to the ratio of the widths. In particular we can see that if $\det \mathbf{H}_B < \det \mathbf{H}_A$, then $Y < 0$, and so no amount of noise will result in $\boldsymbol{\theta}_A$ being more probable than $\boldsymbol{\theta}_B$. In words, if the minimum at $\boldsymbol{\theta}_A$ is both higher and sharper than the minimum at $\boldsymbol{\theta}_B$, it is never reached with higher probability than $\boldsymbol{\theta}_B$, regardless of the amount of noise. However, if $\det \mathbf{H}_B > \det \mathbf{H}_A$ then $Y > 0$, and there is a lower bound on the noise

$$n > \frac{(L_A - L_B)}{2\sigma^2 \log \left( \sqrt{\frac{\det \mathbf{H}_B}{\det \mathbf{H}_A}} \right)} \tag{4}$$

to make $\boldsymbol{\theta}_A$ more probable than $\boldsymbol{\theta}_B$. In words, if the minimum at $\boldsymbol{\theta}_A$ is higher but flatter than the minimum at $\boldsymbol{\theta}_B$, it is favored over $\boldsymbol{\theta}_B$, as long as the noise is large enough, as defined by eq. (4).

To summarize, the presented theory shows that the noise level in SGD (which is defined by the ratio of learning rate to batch size) controls the extent to which optimization favors wider over deeper minima. Increasing the noise by increasing the ratio of learning rate to batch size increases the probability of wider compared to deeper minima. For a discussion on the relative probabilities of critical points that are not strictly minima, see appendix D.

## 4 EXPERIMENTS

### 4.1 IMPACT OF $\frac{\eta}{S}$ ON THE MINIMA SGD FINDS

In this section, we empirically study the impact of learning rate $\eta$ and batch size $S$ on the local minimum that SGD finds. We first focus on a 4-layer Batch Normalized ReLU MLP trained on Fashion-MNIST (Xiao et al., 2017). We study how the noise ratio $n = \frac{\eta}{S}$ leads to minima with different curvatures and validation accuracy. To measure the curvature at a minimum, we compute the norm of its Hessian (a higher norm implies higher sharpness of the minimum) using the finite difference method (Wu et al., 2017). In Figure 1a, we report the norm of the Hessian for local minima obtained by SGD for different $n = \frac{\eta}{S}$, where $\eta \in [5e - 3, 1e - 1]$ and $S \in [25, 1000]$. Each experiment is run for 200 epochs; most models reach approximately $100\%$ accuracy on train

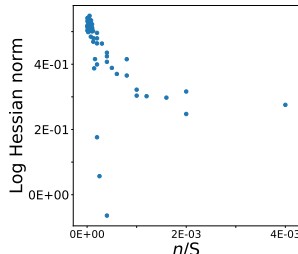 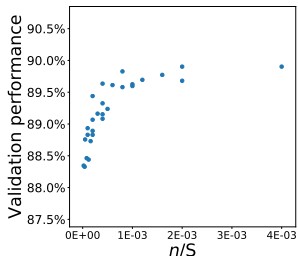

(a) Correlation of $\frac{\eta}{S}$ with logarithm of norm of Hessian.

(b) Correlation of $\frac{\eta}{S}$ with validation accuracy.

Figure 1: Impact on SGD with ratio of learning rate $\eta$ and batch size $S$ for 4 layer ReLU MLP on FashionMNIST.

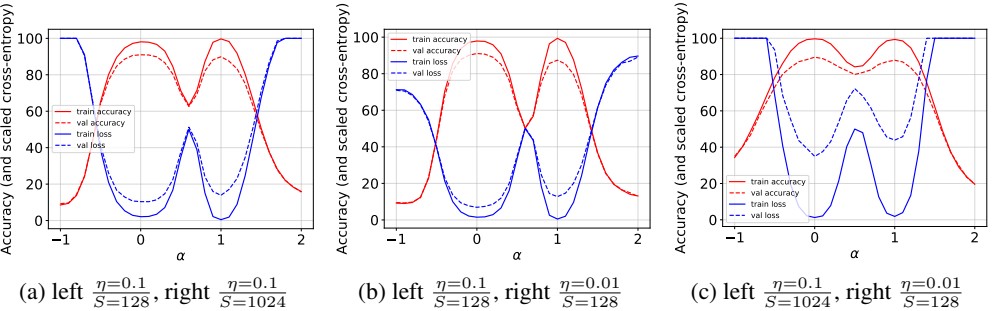

(a) left $\frac{\eta=0.1}{S=128}$, right $\frac{\eta=0.1}{S=1024}$   (b) left $\frac{\eta=0.1}{S=128}$, right $\frac{\eta=0.01}{S=128}$   (c) left $\frac{\eta=0.1}{S=1024}$, right $\frac{\eta=0.01}{S=128}$

Figure 2: Interpolation of Resnet56 networks trained with different learning rate to batch size ratio, $\frac{\eta}{S}$. $\alpha$ (x-axis) corresponds to the interpolation coefficient. As predicted by our theory, lower $\frac{\eta}{S}$ ratio leads to sharper minima (as shown by the left and middle plot).

set. As $n$ grows, we observe that the norm of the Hessian at the minima also decreases, suggesting that higher $\frac{\eta}{S}$ pushes the optimization towards flatter minima. This agrees with Theorem 2, Eq. (3), that higher $\frac{\eta}{S}$ favors flatter over sharper minima.

Figure 1b shows the results from exploring the impact of $n = \frac{\eta}{S}$ on the final validation performance, which confirms that better generalization correlates with higher $n$. Taken together, Fig. 1a and Fig. 1b imply wider minima correlate well with better generalization. As $n = \frac{\eta}{S}$ increases, SGD finds local minima that generalize better. In Appendix F.1, we report similar results for Resnet56 applied on CIFAR10 in Figure 8, for a 20 layer ReLU network with good initialization schemes in Figures 9a and 9c, and with bad initilization in Figure 9b.

To further illustrate the behavior of SGD with different noise levels, we train three Resnet56 models on CIFAR10 using SGD (without momentum) with different $\frac{\eta}{S}$. Our baseline model uses $\frac{\eta=0.1}{S=128}$. In comparision, we investigate a large batch model with $\frac{\eta=0.1}{S=1024}$ and a small learning rate model with $\frac{\eta=0.01}{S=128}$, which have approximately the same $\frac{\eta}{S}$ ratio. We follow (Goodfellow et al., 2014) by investigating the loss on the line interpolating between the parameters of two models. More specifically, let $\boldsymbol{\theta}_1$ and $\boldsymbol{\theta}_2$ be the final parameters found by SGD using different $\frac{\eta}{S}$, we report the loss values $L((1 - \alpha)\boldsymbol{\theta}_1 + \alpha\boldsymbol{\theta}_2)$ for $\alpha \in [-1, 2]$. Results indicate that models with large batch size (Fig. 2-left) or low learning rate (Fig. 2-middle; both having a lower $\frac{\eta}{S}$ than the baseline) end up in a sharper minimum relative to the baseline model. These plots are consistent with our theoretical analysis that higher $n = \eta/S$ gives preference to wider minima over sharper minima. On the other hand, figure 2 (right) shows that models trained with roughly the same level of noise end up in minima of similar quality. The following experiment explores this aspect further.

We train VGG-11 models (Simonyan & Zisserman, 2014) on CIFAR-10, such that all the models are trained with the same noise level but with different values of learning rate and batch size. Specifically, we use $\frac{\eta=0.1\times\beta}{S=50\times\beta}$, where we set $\beta = 0.25, 1, 4$. We then interpolate between the model parameters found when training with $\beta = 1$ and $\beta = 4$ (Fig. 3-left), and $\beta = 1$ and $\beta = 0.25$

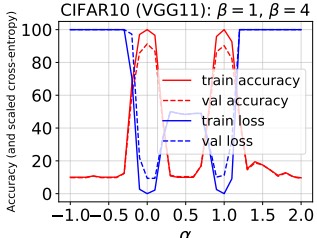 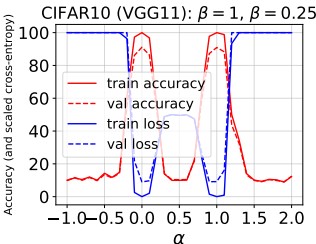

(a) $\beta = 1$ corresponds to model at $\alpha = 0$ and $\beta = 4$ corresponds to model at $\alpha = 1$

(b) $\beta = 1$ corresponds to model at $\alpha = 0$ and $\beta = 0.25$ corresponds to model at $\alpha = 1$

Figure 3: Interpolation between parameters of models trained with the same learning rate ($\eta$) to batch-size ($S$) ratio: $\frac{\eta = 0.1 \times \beta}{S = 50 \times \beta}$, but different $\eta$ and $S$ values determined by $\beta$. As predicted by our theory, the minima for models with identical noise levels should be qualitatively similar as can be seen by these plots.

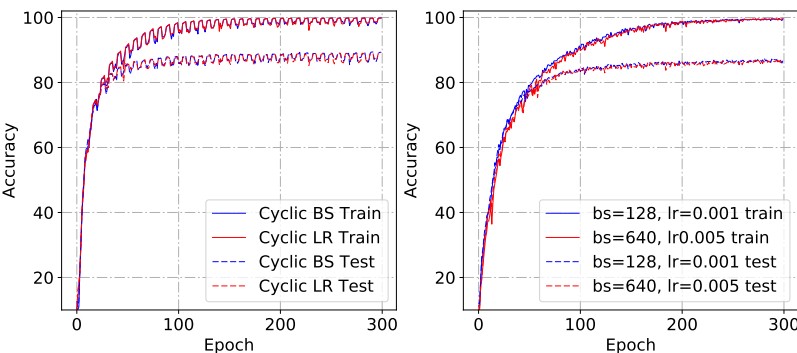

Figure 4: Learning rate schedule can be replaced by an equivalent batch size schedule. The ratio of learning rate to batch size is equal at all times for both red and blue curves in each plot. Above plots show train and test accuracy for experiments involving VGG-11 architecture on CIFAR10 dataset. Left: cyclic batch size schedule (blue) in range 128 to 640, compared to cyclic learning rate schedule (red) in range 0.001 to 0.005. Right: constant batch size 128 and constant learning rate 0.001 (blue), compared to constant batch size 640 and constant learning rate 0.005 (red).

(Fig. 3-right). The interpolation results indicate that all the minima have similar width and depth, qualitatively supporting our theoretical observation that for the same noise ratio SGD ends up in minima of similar quality.

## 4.2  $\frac{\eta}{S}$ RATIO INFLUENCES LEARNING DYNAMICS OF SGD

In this section we look at two experimental phenomena: firstly, the equilibrium endpoint of SGD and secondly the dynamical evolution of SGD. The former, was theoretically analysed in the theory section, while the latter is not directly addressed in the theory section, but we note that the two are related - the endpoint is the result of the intermediate dynamics.

We experimentally study both phenomena in the following four experiments involving the VGG-11 architecture on the CIFAR10 dataset, shown in Fig 4. The left plot compares two experiments: cyclic batch size schedule (blue) in range 128 to 640, compared to cyclic learning rate schedule (red) in range 0.001 to 0.005. The right plot compares two other experiments: constant learning rate to batch-size ratio of $\frac{\eta}{S} = \frac{0.001}{128}$ (blue) and $\frac{\eta}{S} = \frac{0.005}{640}$ (red).

Regarding the first phenomena, of the endpoint of SGD, the test accuracy when training with a cyclic batch size and cyclic learning rate is 89.39% and 89.24%, respectively, and we emphasize that these are similar scores. For a constant learning rate to batch-size ratio of $\frac{\eta}{S} = \frac{0.001}{128}$ and $\frac{\eta}{S} = \frac{0.005}{640}$, the

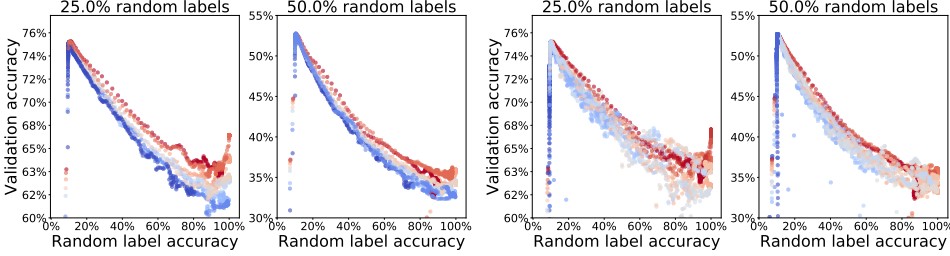

Figure 5: Impact of $\frac{\eta}{S}$ on memorization of MNIST when $25\%$ and $50\%$ of labels in the training set are replaced with random labels, using no momentum (on the right) or a momentum with parameter $0.9$ (on the left). We observe that for a specific level of memorization, high $\frac{\eta}{S}$ leads to better generalization. Red has higher value of $\frac{\eta}{S}$ than blue.

test accuracy is $87.25\%$ and $86.92\%$, respectively, and we again emphasize that these two scores are similar to each other. That in each of these experiments the endpoint test accuracies are similar shows exchangability of learning rate for batch size for the endpoint, and is consistent with our theoretical calculation which says characteristics of the minima found at the endpoint are determined by the ratio of learning rate to batch-size, but not individually on learning rate or batch size. Additional results exploring cyclical learning rate and batch size schedule are reported in Appendix F.4.

Regarding the second phenomena of the dynamical evolution, we note the similarity of the training and test accuracy curves for each pair of same-noise curves in each experiment. Our theoretical analysis does not explain this phenomena, as it does not determine the dynamical distribution. Nonetheless, we report it here as an interesting observation, and point to Appendix B for some intuition on why this may occur from the Fokker-Planck equation. In Appendix F.2, Fig. 13 we show in more detail the loss curves. While the epoch-averaged loss curves match well when exchanging batch size for learning rate, the per-iteration loss is not invariant to switching batch size for learning rate. In particular, we note that each run with smaller batch-size has higher variance in per-iteration loss than it's same-noise pair. This is expected, since from one iteration to the next, the examples will have higher variance for a smaller batch-size.

The take-away message from this section is that the endpoint and dynamics of SGD are approximately invariant if the batch size and learning rate are simultaneously rescaled by the same amount. This is in contrast to a commonly used heuristic consisting of scaling the learning rate with the square root of the batch size, i.e. of keeping the ratio $\eta/\sqrt{S}$ constant. This is used for example by (Hoffer et al., 2017) as a way of keeping the covariance matrix of the parameter update step the same for any batch size. However, our theory and experiments suggest changing learning rate and batch size in a way that keeps the ratio $n = \eta/S$ constant instead, since this results in the same equilibrium distribution.

### 4.3 IMPACT OF SGD ON MEMORIZATION

To generalize well, a model must identify the underlying pattern in the data instead of simply perfectly memorizing each training example. An empirical approach to test for memorization is to analyze how good a DNN can fit a training set when the true labels are partly replaced by random labels (Zhang et al., 2016; Arpit et al., 2017). The experiments described in this section highlight that SGD with a sufficient amount of noise improves generalization at a given level of memorization.

Experiments are performed on the MNIST dataset with an MLP similar to the one used by (Arpit et al., 2017), but with 256 hidden units. We train the MLP with different amounts of random labels in the training set. For each level of label noise, we evaluate the impact of $\frac{\eta}{S}$ on the generalization performance. Specifically, we run experiments with $\frac{\eta}{S}$ taking values in a grid with batch size in $\{50, 100, 200, 500, 1000\}$, learning rate in $\{0.005, 0.01, 0.02, 0.05, 0.07, 0.1\}$, and momentum in $\{0.0, 0.9\}$. Models are trained for 300 epochs. Fig. 5 reports the MLPs performances on both the noisy training set and the validation set. The results show that larger noise in SGD (regardless if induced by using a smaller batch size or a larger learning rate) leads to solutions which generalize better for the same amount of random labels memorized on the training set. Thus, our analysis

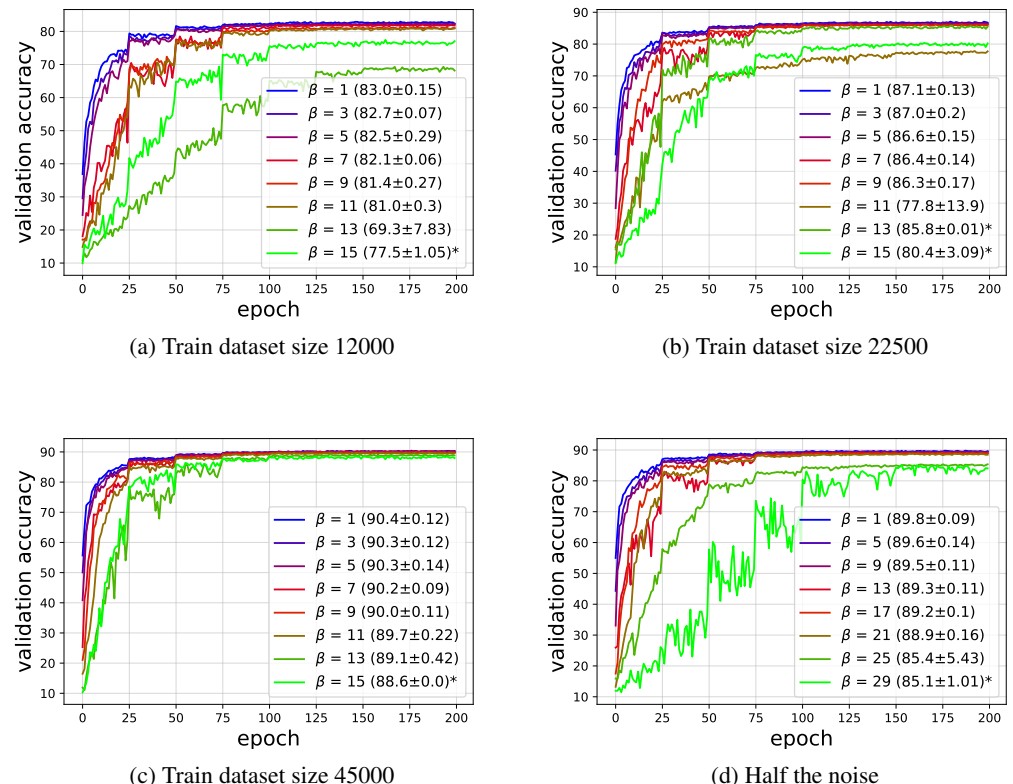

Figure 6: Breaking point analysis: Our theory suggests the final performance should be similar when the SGD noise level $\frac{\beta \times \eta}{\beta \times S}$ is kept the same. Here we study its breaking point in terms of too large a learning rate or too small a batch size. (a,b,c)- Validation accuracy for different dataset sizes and different $\beta$ values for a VGG-11 architecture trained on CIFAR10. In each experiment, we multiply the learning rate ($\eta$) and batch size ($S$) with $\beta$ such that the ratio $\frac{\beta \times (\eta=0.1)}{\beta \times (S=50)}$ is fixed. We observe that for the same ratio, increasing the learning rate and batch size yields similar performance up to a certain $\beta$ value, for which the performance drops significantly. (d)- Breaking point analysis when half the noise level $\frac{\beta \times (\eta=0.05)}{\beta \times (S=50)}$ is used. The breaking point happens for much larger $\beta$ when using a smaller noise. All experiments are repeated 5 times with different random seeds. The graphs denote the mean validation accuracies and the numbers in the brackets denote the mean and standard deviation of the maximum validation accuracy across different runs. The * denotes at least one seed lead to divergence.

highlights that SGD with low noise $n = \frac{\eta}{S}$ steers the endpoint of optimization towards a minimum with low generalization ability.

While Fig 5 reports the generalization at the endpoint, we observe that SGD with larger noise continuously steers away from sharp solutions throughout the dynamics. We also reproduce the observation reported by (Arpit et al., 2017): that memorization roughly starts after reaching maximum generalization. For runs with momentum we exclude learning rates higher than 0.02 as they lead to divergence. Full learning curves are reported in Fig. 14 included in Appendix F.3.

## 4.4 BREAKING POINT OF THE THEORY IN PRACTICE

Our analysis relies on the assumption that the gradient step is sufficiently small to guarantee that the first order approximation of a Taylor's expansion is a good estimate of the loss function. In the case where the learning rate becomes too high, this approximation is no longer suitable, and the continuous limit of the discrete SGD update equation will no longer be valid. In this case, the

stochastic differential equation doesn't hold, and hence neither does the Fokker-Planck equation, and so we don't expect our theory to be valid. In particular, we don't expect to arrive at the same stationary distribution as indicated by a fixed ratio $\eta/S$, if the learning rate gets too high.

This is exemplified by the empirical results reported in Fig. 6, where similar learning dynamics and final performance can be observed when simultaneously multiplying the learning rate and batch size by a factor $\beta$ up to a certain limit. This is done for different training set sizes to investigate if the breaking point depends on this factor (Fig. 6 a,b,c). The plots suggest that the breaking point happens for smaller $\beta$ values if the dataset size is smaller. We also investigate the influence of $\beta$ when half the noise level is used, due to halving the learning rate, in (figure 6 d). These experiments strongly suggest that the reason behind breaking point is the use of a high learning rate because the performance drops at much higher $\beta$ when the base learning rate is halved. A similar experiment is performed on Resnets (for results see Fig 7 in the appendix). We highlight other limitations of our theory in appendix E.

## 5 DISCUSSION

In the theoretical section of this work we treat the learning rate as fixed throughout training. However, in practical applications, the learning rate is annealed to a lower value, either gradually or in discrete jumps. When viewed within our framework, at the beginning with high noise, SGD favors width over depth of a region, then as the noise decreases, SGD prioritizes the depth more strongly - this can be seen from Theorem 3 and the comments that follow.

In the theoretical section we made the additional assumption that the covariance of the gradients is isotropic, in order to be able to derive a closed form solution for the equilibrium distribution. We do not expect this assumption to hold in practice, but speculate that there may be mechanisms which drive the covariance towards isotropy, for example one may be able to tune learning rates on a per-parameter basis in such a way that the combination of learning rate and covariance matrix is approximately isotropic – this may lead to improvements in optimization. Perhaps some existing mechanisms such as batch normalization or careful initialization give rise to more equalized covariance - we leave study of this for future work.

We note further that our theoretical analysis considered an equilibrium distribution, which was independent of the intermediate dynamics. However, this may not be the case in practice. Without the isotropic covariance, the system of partial differential equations in the late time limit will in general have a solution which will depend on the path through which optimization occurs, unless other restrictive assumptions are made to force this path dependence to disappear (Chaudhari & Soatto, 2017). Despite this simplifying assumption, our empirical results are consistent with the developed theory. We leave study of path dependence and dynamics to future work.

In experiments investigating memorization we explored how the noise level changes the preference of wide minima over sharp ones. (Arpit et al., 2017) argues that SGD first learns true labels, before focusing on random labels. Our insight is that in the second phase the high level of noise maintains generalization. This illustrates the trade-off between width of minima and depth in practice. When the noise level is lower, DNNs are more likely to fit random labels better, at the expense of generalizing less well on true ones.

## 6 CONCLUSIONS

We shed light on the role of noise in SGD optimization of DNNs and argue that three factors (batch size, learning rate and gradient variance) strongly influence the properties (loss and width) of the final minima at which SGD converges. The learning rate and batch size of SGD can be viewed as one effective hyper-parameter acting as a noise factor $n = \eta/S$. This, together with the gradient covariance influences the trade-off between the loss and width of the final minima. Specifically, higher noise favors wider minima, which in turn correlates with better generalization.

Further, we experimentally verify that the noise $n = \eta/S$ determines the width and height of the minima towards which SGD converges. We also show the impact of this noise on the memorization phenomenon. We discuss the limitations of the theory in practice, exemplified by when the learning

rate gets too large. We also experimentally verify that $\eta$ and $S$ can be simultaneously rescaled as long as the noise $\eta/S$ remains the same.

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

APPENDIX

## A  DERIVATION OF THE FOKKER-PLANCK EQUATION

In this appendix we derive the Fokker-Planck equation. For any stochastic differential equation, since the evolution is noisy we can't say exactly where in parameter space the parameter values will be at any given time. But we can talk about the probability $P(\boldsymbol{\theta}, t|\boldsymbol{\theta}_0, t_0)$ that a parameter takes a certain value $\boldsymbol{\theta}$ at a certain time $t$, given that it started at $\boldsymbol{\theta}_0, t_0$. That is captured by the Fokker-Planck equation, which reads

$$\frac{\partial P(\boldsymbol{\theta}, t)}{\partial t} = \nabla_{\boldsymbol{\theta}} \cdot \left[ \eta \boldsymbol{g}(\boldsymbol{\theta}) P(\boldsymbol{\theta}, t) + \frac{\eta^2}{2S} \nabla_{\boldsymbol{\theta}} \cdot [\mathbf{C}(\boldsymbol{\theta}) P(\boldsymbol{\theta}, t)] \right]. \tag{5}$$

In this appendix we will derive the above equation from the stochastic differential equation (6). We will not be interested in pure mathematical rigour, but intend the proof to add intuition for the machine learning audience.

For brevity we will sometimes write the probability just as $P(\boldsymbol{\theta}, t)$. We will sometimes make use of tensor index notation, where a tensor is denoted by its components, (for example $\theta_i$ are the components of the vector $\boldsymbol{\theta}$), and we will use the index summation convention, where a repeated index is to be summed over.

We start with the stochastic differential equation

$$\frac{d\boldsymbol{\theta}}{dt} = -\eta \boldsymbol{g}(\boldsymbol{\theta}) + \frac{\eta}{\sqrt{S}} \mathbf{B}(\boldsymbol{\theta}) \mathbf{f}(t). \tag{6}$$

A formal expression for the probability for a given noise function $\mathbf{f}$ is given by $P(\boldsymbol{\theta}, t) = \delta(\boldsymbol{\theta} - \boldsymbol{\theta}_{\mathbf{f}})$, but since we don't know the noise, we instead average over all possible noise functions to get

$$P(\boldsymbol{\theta}, t) = \mathbb{E}[\delta(\boldsymbol{\theta} - \boldsymbol{\theta}_{\mathbf{f}})]. \tag{7}$$

While this is just a formal solution we will make use of it later in the derivation.

We now consider a small step in time $\delta t$, working at first order in $\delta t$, and ask how far the parameters move, denoted $\delta \boldsymbol{\theta}$, which is given by integrating (6)

$$\delta \boldsymbol{\theta} = -\eta \boldsymbol{g} \delta t + \frac{\eta}{\sqrt{S}} \mathbf{B} \int_t^{t+\delta t} \mathbf{f}(t') dt' \tag{8}$$

where we've assumed that $\delta \boldsymbol{\theta}$ is small enough that we can evaluate $\boldsymbol{g}$ at the original $\boldsymbol{\theta}$. We now look at expectations of $\delta \boldsymbol{\theta}$. Using the fact that the noise is normalized Gaussian, $\mathbb{E}(\mathbf{f}(t)) = 0$, we get (switching to index notation for clarity)

$$\mathbb{E}(\delta \theta_i) = -\eta g_i \delta t \tag{9}$$

and using that the noise is normalized Gaussian again, we have that $\mathbb{E}(\mathbf{f}(t)\mathbf{f}(t')) = \mathbf{I}\delta(t - t')$, leading to

$$\mathbb{E}(\delta \theta_i \delta \theta_j) = \frac{\eta^2}{S} C_{ij} \delta t + \mathcal{O}(\delta t^2). \tag{10}$$

If at time $t$ we are at position $\boldsymbol{\theta}'$ and end up at position $\boldsymbol{\theta} = \boldsymbol{\theta}' + \delta\boldsymbol{\theta}$ at time $t + \delta t$, then we can take (7) with $\boldsymbol{\theta_f} = \boldsymbol{\theta}'$ and Taylor expand it in $\delta\theta_i$

$$P(\boldsymbol{\theta}, t + \delta t | \boldsymbol{\theta}', t) = \left( 1 + \mathbb{E}(\delta\theta_i)\frac{\partial}{\partial\theta_i'} + \frac{1}{2}\mathbb{E}(\delta\theta_i\delta\theta_j)\frac{\partial^2}{\partial\theta_i'\partial\theta_j'} + \mathcal{O}(\delta t^2) \right) \delta(\boldsymbol{\theta} - \boldsymbol{\theta}'). \quad (11)$$

We deal with the derivatives of the delta functions in the following way. We will use the following identity, called the Chapman-Kolmogorov equation, which reads

$$P(\boldsymbol{\theta}, t + \delta t | \boldsymbol{\theta}_0, t_0) = \int_{-\infty}^{+\infty} d\boldsymbol{\theta}' P(\boldsymbol{\theta}, t + \delta t | \boldsymbol{\theta}', t') P(\boldsymbol{\theta}', t' | \boldsymbol{\theta}_0, t_0) \quad (12)$$

for any $t'$, such that $t_0 \leq t' \leq t + \delta t$. This identity is an integral version of the chain rule of probability, stating there are multiple paths of getting from an initial position $\boldsymbol{\theta}_0$ to $\boldsymbol{\theta}$ at time $t + \delta t$ and one should sum all of these paths.

We will now substitute (11) into the first term on the right hand side of (12) with $t'$ set to be $t$, and apply integration by parts (assuming vanishing boundary conditions at infinity) to move the derivatives off of the delta functions and onto the other terms. We end up with

$$P(\boldsymbol{\theta}, t + \delta t | \boldsymbol{\theta}', t) = P(\boldsymbol{\theta}, t | \boldsymbol{\theta}_0, t_0) - \frac{\partial}{\partial\theta_i} \left( \mathbb{E}(\delta\theta_i) P(\boldsymbol{\theta}, t | \boldsymbol{\theta}_0, t_0) \right)$$
$$+ \frac{1}{2}\frac{\partial^2}{\partial\theta_i'\partial\theta_j'} \left( \mathbb{E}(\delta\theta_i\delta\theta_j) P(\boldsymbol{\theta}, t | \boldsymbol{\theta}_0, t_0) \right). \quad (13)$$

We can then take the first term on the right hand side of (13) to the other side, insert (9) and (10), divide by $\delta t$ and take the limit $\delta t \to 0$, getting a partial derivative with respect to time on the left hand side, leading directly to the Fokker-Planck equation quoted in the text (5) (where we have reverted back to vector notation from index notation for conciseness)

$$\frac{\partial P(\boldsymbol{\theta}, t)}{\partial t} = \nabla_{\boldsymbol{\theta}} \cdot \left[ \eta\boldsymbol{g}(\boldsymbol{\theta})P(\boldsymbol{\theta}, t) + \frac{\eta^2}{2S}\nabla_{\boldsymbol{\theta}} \cdot [\mathbf{C}(\boldsymbol{\theta})P(\boldsymbol{\theta}, t)] \right]. \quad (14)$$

## B    INTUITIONS FROM FOKKER-PLANCK

In this appendix we will give some supplementary comments about the intuition we can gain from the Fokker-Planck equation (5). If the learning rate, batch size are constant and the covariance is proportional to the identity $\mathbf{C} = \sigma^2\mathbf{I}$ and $\sigma^2$ is constant, then we can rewrite the Fokker-Planck equation in the following form

$$\frac{\partial P(\boldsymbol{\theta}, t_\eta)}{\partial t_\eta} = \nabla_{\boldsymbol{\theta}} \cdot [\mathbf{g}(\boldsymbol{\theta})P(\boldsymbol{\theta}, t_\eta)] + \frac{\eta\sigma^2}{2S}\nabla_{\boldsymbol{\theta}}^2 P(\boldsymbol{\theta}, t_\eta), \quad (15)$$

where we have rescaled the time coordinate to be $t_\eta \equiv t\eta$. One can now see that in terms of this rescaled time coordinate, the ratio between the drift and diffusion terms is governed by the following ratio

$$\frac{\eta\sigma^2}{2S}. \quad (16)$$

In terms of the balance between drift and diffusion, we see that a higher value of this ratio gives rise to a more diffusive evolution, while a lower value allows the potential drift term to dominate. In the next section we will see how this ratio controls the stationary distribution at which SGD converges to. For now we highlight that in terms of this rescaled coordinate, only this ratio controls the evolution towards this stationary distribution (not just its endpoint), in terms of the rescaled time $t_\eta$. That is, learning rate and batch size are interchangable in the sense that the ratio is invariant under transformations $S \to aS$, $\eta \to a\eta$, for $a > 0$. But note that the time it takes to reach the stationary distribution depends on $\eta$ as well, because of the rescaled time variable. For example, for a higher learning rate, but constant ratio, one arrives at the same stationary distribution, but in a quicker time by a factor of $1/\eta$.

However, a caution here is necessary. The first order SGD update equation only holds for small enough $\eta$ that a first order approximation to a Taylor expansion is valid, and hence we expect the first order approximation of SGD as a continuous stochastic differential equation to break down for high $\eta$. Thus, we expect learning rate and batch size to be interchangable up to a maximum value of $\eta$ at which the approximation breaks.

## C    Equilibrium Distribution

In this appendix we will prove equation (2) quoted in the main text, which claims that for an isotropic covariance, $\mathbf{C}(\boldsymbol{\theta}) = \sigma^2 \mathbf{I}$, with $\sigma$ constant, the equilibrium solution of the Fokker-Planck equation has the following form

$$P(\boldsymbol{\theta}) = P_0 \exp\left(-\frac{2L(\boldsymbol{\theta})}{n\sigma^2}\right) \quad , \tag{17}$$

where $n \equiv \frac{\eta}{S}$ and $P_0$ is a normalization constant, which is well defined for loss functions with $L_2$ regularization.

In order to prove this is the equilibrium distribution, we need to solve the Fokker-Planck equation (5) with the left hand side set equal to zero (which would give a stationary distribution) and further we require for equilibrium that detailed balance holds. To do this, we begin by writing the Fokker-Planck equation in a slightly different form, making use of a probability current $\boldsymbol{J}$, defined

$$\boldsymbol{J} \equiv \eta \boldsymbol{g}(\boldsymbol{\theta}) P(\boldsymbol{\theta}, t) + \frac{\eta^2}{2S} \nabla_{\boldsymbol{\theta}} \cdot (\mathbf{C}(\boldsymbol{\theta}) P(\boldsymbol{\theta}, t)) \tag{18}$$

in which the Fokker-Planck equation becomes

$$\frac{\partial P(\boldsymbol{\theta}, t)}{\partial t} = \nabla_{\boldsymbol{\theta}} \cdot \boldsymbol{J}. \tag{19}$$

At this point we use the assumption that $\mathbf{C}(\boldsymbol{\theta}) = \sigma^2 \mathbf{I}$ to get

$$\boldsymbol{J} = \eta \boldsymbol{g}(\boldsymbol{\theta}) P(\boldsymbol{\theta}, t) + \frac{\eta^2 \sigma^2}{2S} \nabla_{\boldsymbol{\theta}} P(\boldsymbol{\theta}, t). \tag{20}$$

The stationary solution has $\frac{\partial P(\boldsymbol{\theta}, t)}{\partial t} = 0$ and hence $\nabla_{\boldsymbol{\theta}} \cdot \boldsymbol{J} = 0$. But we require the equilibrium solution, which is a stronger demand than just the stationary solution. The equilibrium solution is a particular stationary solution in which detailed balance occurs. Detailed balance means that in the stationary solution, each individual transition balances precisely with its time reverse, resulting in zero probability currents, see §5.3.5 of (Gardiner, 2010), i.e. that $\boldsymbol{J} = 0$. Detailed balance is a sufficient condition for having entropy increasing with time. Non-zero $\boldsymbol{J}$ with $\nabla_{\boldsymbol{\theta}} \cdot \boldsymbol{J} = 0$ would correspond to a non-equilibrium stationary distribution, which we don't consider here.

For the equilibrium solution, $\boldsymbol{J} = 0$, we get the result for the probability distribution

$$P(\boldsymbol{\theta}) = P_0 \exp\left(-\frac{2S}{\eta\sigma^2} L(\boldsymbol{\theta})\right). \tag{21}$$

which is the desired stationary solution we intended to find. Finally, to ensure that $P_0$ is a finite value, we note that the loss function $L(\boldsymbol{\theta})$ can be decomposed as,

$$L(\boldsymbol{\theta}) = L_0(\boldsymbol{\theta}) + \tau \|\boldsymbol{\theta}\|^2 \tag{22}$$

where $\tau$ is some non-negative constant controlling $L_2$ regularization. Then we see that,

$$P_0 = \int_{\boldsymbol{\theta}} \exp\left(-\frac{2S}{\eta\sigma^2} L(\boldsymbol{\theta})\right) \tag{23}$$

$$= \int_{\boldsymbol{\theta}} \exp\left(-\frac{2S}{\eta\sigma^2} L_0(\boldsymbol{\theta}) - \tau \|\boldsymbol{\theta}\|^2\right) \tag{24}$$

$$= \int_{\boldsymbol{\theta}} \exp\left(-\frac{2S}{\eta\sigma^2} L_0(\boldsymbol{\theta})\right) \exp\left(-\tau \|\boldsymbol{\theta}\|^2\right) \tag{25}$$

Since $L_0(\boldsymbol{\theta}) \geq 0$ and $\frac{2S}{\eta\sigma^2} > 0$, we have that $\exp\left(-\frac{2S}{\eta\sigma^2} L_0(\boldsymbol{\theta})\right) \leq 1$, thus,

$$\int_{\boldsymbol{\theta}} \exp\left(-\frac{2S}{\eta\sigma^2} L_0(\boldsymbol{\theta})\right) \exp\left(-\tau \|\boldsymbol{\theta}\|^2\right) \leq \int_{\boldsymbol{\theta}} \exp\left(-\tau \|\boldsymbol{\theta}\|^2\right) \tag{26}$$

and we now note that the right hand side of (26) is finite because it is just a multidimensional Gaussian integral. Thus, $P_0$ has a finite value and hence the stationary distribution $P(\boldsymbol{\theta})$ is well defined.

## D   DERIVATION OF PROBABILITY OF ENDING IN A GIVEN MINIMA

In this appendix we derive the discrete set of probabilities of ending at each minima, as given in (3). Essentially, we will use Laplace's method to approximate the integral of the probability density in the region near a minimum. This is a common approach used to approximate integrals, appearing for example in (MacKay, 1992a) and (Kass & Raftery, 1995). We work locally near $\boldsymbol{\theta}_A$ and take the following approximation of the loss function, since it is near a minimum,

$$L(\boldsymbol{\theta})\Big|_{R_A} \approx L_A + \frac{1}{2}(\boldsymbol{\theta} - \boldsymbol{\theta}_A)^\top \mathbf{H}_A(\boldsymbol{\theta} - \boldsymbol{\theta}_A). \tag{27}$$

where $\mathbf{H}_A$ is the Hessian, and is positive definite for a minimum, and where the subscript $R_A$ indicates that this approximation only holds in the region $R_A$ near the minimum $\boldsymbol{\theta}_A$. We emphasize this is not the same as Assumption 4 of (Mandt et al., 2017), where they assume the loss can be globally approximated by a quadratic. In contrast, we allow for a general loss, with multiple minima, and locally to each minima approximate the loss by its second-order Taylor expansion in order to evaluate the integral in the Laplace method.

The distribution $P_A(\boldsymbol{\theta})$ is a probability density, while we are interested in the discrete set of probabilities of ending at a given minimum, which we will denote by lowercase $p_A$. To calculate this discrete set of probabilities, for each minimum we need to integrate this stationary distribution over an interval containing the minimum.

Integrating (17) in some region $R_A$ around $\boldsymbol{\theta}_A$ and using (27)

$$p_A \approx P_0 \int_{R_A} \exp\left(-\frac{2S}{\eta\sigma^2} L(\boldsymbol{\theta})\Big|_{R_A}\right) \tag{28}$$

$$\approx P_0 \int_{R_A} \exp\left(-\frac{2S}{\eta\sigma^2}\left[L_A + \frac{1}{2}(\boldsymbol{\theta} - \boldsymbol{\theta}_A)^\top \mathbf{H}_A(\boldsymbol{\theta} - \boldsymbol{\theta}_A)\right]\right) \tag{29}$$

$$\approx \tilde{P}_0 \exp\left(-\frac{2SL_A}{\eta\sigma^2}\right) \sqrt{\frac{1}{\det \mathbf{H}_A}} \tag{30}$$

where in the last line we assume the region is large enough that an approximation to the full Gaussian integral can be used (i.e. that the tails don't contribute, which is fine as long as the region is sufficiently larger than $\det \mathbf{H}_A$) – note that the region can't be too large, otherwise we would invalidate our local assumption. The picture is that the minima are sufficiently far apart that the region can be taken sufficiently large for this approximation to be valid. Note that $\tilde{P}_0$ is different to $P_0$ and includes the normalization factors from performing the Gaussian integral, which are the same for all minima.

Since we are interested in relative probabilities between different minima, so we can consider the unnormalized probability, dropping factors that are common amongst all minima we get

$$\tilde{p}_A = \exp\left(-\frac{2SL_A}{\eta\sigma^2}\right) \sqrt{\frac{1}{\det \mathbf{H}_A}} \tag{31}$$

This is the required expression given in the main text.

We note that the derivation used above talks about strict minima, i.e., minima with a positive definite Hessian. In practice however, for deep neural networks with a large number of parameters, it is unrealistic to expect the endpoint of training to be in a strict minimum. Instead, it is more likely to be a point at which the Hessian has positive eigenvalues in a few directions, while the other eigenvalues are (approximately) zero. In such cases, to understand which minima SGD favors, we can consider the fact that at equilibrium, the iterate of SGD follows the distribution,

$$P(\boldsymbol{\theta}) = P_0 \exp\left(-\frac{2L(\boldsymbol{\theta})}{n\sigma^2}\right) \tag{32}$$

By definition this means that at any time during equilibrium, the iterate is more likely to be found in a region of higher probability. In the restrictive case of strict minimum, we model it as a Gaussian and characterize the probability of landing in a minimum depending on the curvature and depth of the loss around that minimum. In the general case of minima with degenerate (flat) directions, we can say that a minimum with more volume is a more probable one.

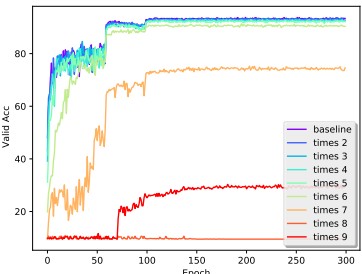

Figure 7: Experiments involving Resnet56 architecture on CIFAR10 dataset. In each curve, we multiply the $\frac{\eta}{S}$ ratio by a given factor (increasing both batch size and learning rate). We observe that multiplying the ratio by a factor up to 5 results in similar performances. However, the performance degrades for factor superior to 5.

# E    LIMITATIONS OF OUR ANALYSIS

Due to our assumptions, we expect the theory to become unreliable when the discrete to continuous approximation fails, when the covariance in gradients is non-isotropic, when the batch size becomes comparable to the finite size of the training set and when momentum is considered.

We discussed the limits of the discrete to continous approximation in 4.4 (further illustrated in 7).

When the covariance in gradients is highly non-isotropic, the equilibrium solution to the Fokker-Planck equation is the solution to a complicated partial differential equation, and one can't easily spot the solution by inspection as we do in Appendix C. We expect this approximation to break down especially in the case of complicated architectures where different gradient directions will be have very different gradient covariances.

Our theory does not involve the finite size of the training set, which is a drawback of the theory. This may be especially apparent when the batch size becomes large, compared to the training set size, and we expect the theory to break down at this point.

Finally, we mention that momentum is used in practical deep learning optimization algorithms. Our theory does not consider momentum, which is a drawback of the theory and we expect it to break down in models in which momentum is important. We can write a Langevin equation for the case of momentum, with momentum damping coefficient $\mu$,

$$\frac{d\boldsymbol{v}}{dt} = -\mu\boldsymbol{v} - \eta\boldsymbol{g} + \frac{\eta\boldsymbol{B}}{\sqrt{S}}\boldsymbol{f}(t)$$

where the velocity is $\boldsymbol{v} = d\boldsymbol{\theta}/dt$.

The Fokker-Planck equation corresponding to this Langevin equation with momentum is

$$\left(\frac{\partial}{\partial t} + \boldsymbol{v} \cdot \nabla_{\boldsymbol{\theta}}\right) P = \eta\nabla_{\boldsymbol{\theta}} \cdot \left(\frac{\mu}{\eta}\boldsymbol{v}P + \boldsymbol{g}P\right) + \frac{\eta^2\sigma^2}{2S}\nabla_{\boldsymbol{\theta}}^2 P$$

where $P$ is now a function of velocity $\boldsymbol{v}$ as well as $\boldsymbol{\theta}$ and $t$, and we've again assumed the gradient covariance varies slowly compared to the loss.

From this more complicated Fokker-Planck equation, it is hard to spot the equilibrium distribution. We leave the study of this for further work. For now, we just note that a factor of $\eta$ can be taken from the right hand side, and then the ratio between the diffusion and drift terms will again be the ratio $\eta\sigma^2/(2S)$.

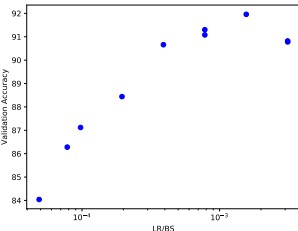

Figure 8: Validation accuracy of Resnet56 networks against different ratios learning rate to batch size on CIFAR10. Trained with SGD.

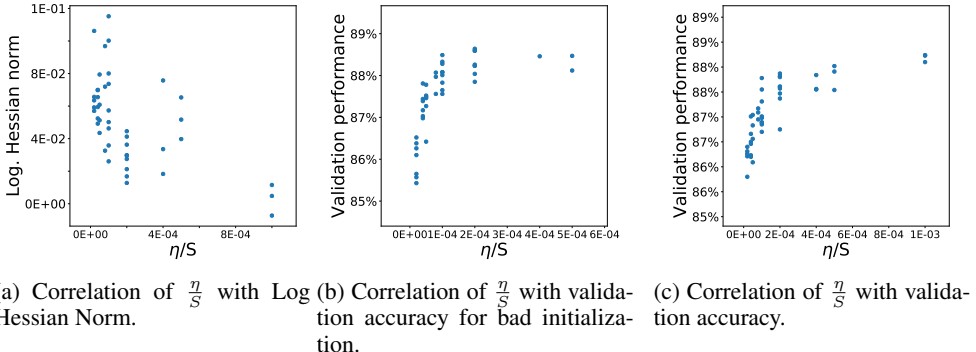

(a) Correlation of $\frac{\eta}{S}$ with Log Hessian Norm.

(b) Correlation of $\frac{\eta}{S}$ with validation accuracy for bad initialization.

(c) Correlation of $\frac{\eta}{S}$ with validation accuracy.

Figure 9: Impact on SGD with ratio of learning rate $\eta$ and batch size $S$ for 20 layer ReLU MLP without Batch Normalization on FashionMNIST.

## F    OTHER EMPIRICAL RESULTS

### F.1    FURTHER IMPACTS OF SGD ON MINIMA

In this appendix we look at other experiments exploring the correlation between learning rate to batch-size ratio and sharpness of minima, and to validation performance. In Figure 8, we report the validation accuracy for Resnet56 models trained on CIFAR10 with different learning rate to batch-size ratio. We notice there is a peak validation accuracy at a learning rate to batch-size ratio of around $2 \times 10^{-3}$. In particular, this example emphasizes that higher learning rate to batch-size ratio doesn't necessarily lead to higher validation accuracy. Instead, it acts as a control on the validation accuracy, and there is an optimal learning rate to batch-size ratio.

Figures 9a and 9c show the results of a variant of 4 layer ReLU experiment, where we use 20 layers and remove Batch Normalization. The inspiration is to test predictions of theory in the more challenging setup. We again observe a correlation of Hessian norm with learning rate to batch-size ratio (though weaker), and similarly between validation performance and learning rate to batch-size ratio.

Figures 10,11, 12 report additional line interpolation plots between models that uses different learning rate to batch size ratio but similar learning rate decay throught training. We repeat the experiment several time to ensure robustness with respect to the model random initializations. Results indicate that models with large batch size or low learning rate end up in a sharper minimum relative to the baseline model.

### F.2    MORE LEARNING RATE & BATCH SIZE EXCHANGEABILITY

In this appendix we show in more detail the experiments of Section 4.2 which show the exchangeability of learning rate and batch size. In Fig. 13 we show the log cross entropy loss, the epoch-averaged log cross entropy loss, train, validation and test accuracy as well as the batch size schedule, learning rate schedule. We see in the learning rate to batch-size ratio plot that the orange and blue

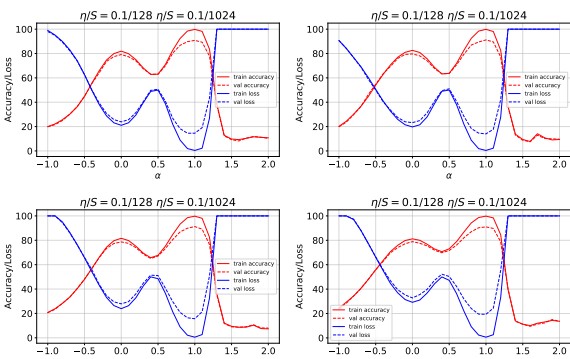

Figure 10: Interpolation of Resnet56 networks trained with different learning rate to batch size ratio, $\frac{\eta}{S}$. $\alpha$ (x-axis) corresponds to the interpolation coefficient. Each plots correspond to model using a different random initialization.

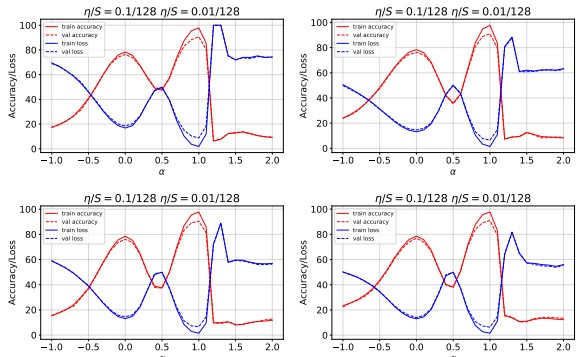

Figure 11: Interpolation of Resnet56 networks trained with different learning rate to batch size ratio, $\frac{\eta}{S}$. $\alpha$ (x-axis) corresponds to the interpolation coefficient. Each plots correspond to model using a different random initialization.

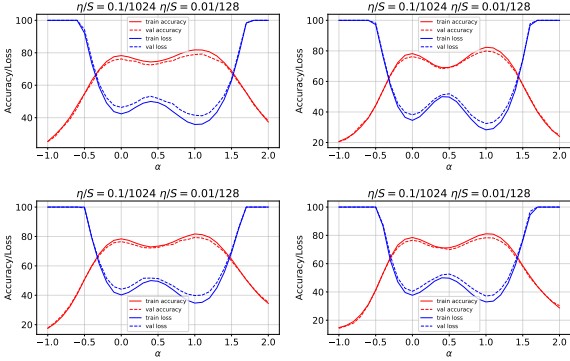

Figure 12: Interpolation of Resnet56 networks trained with different learning rate to batch size ratio, $\frac{\eta}{S}$. $\alpha$ (x-axis) corresponds to the interpolation coefficient. Each plots correspond to model using a different random initialization.

|  | Test acc | Valid acc | Loss | Hessian norm. |
|---|---|---|---|---|
| Discrete $\eta$ | $90.04\% \pm 0.18\%$ | $90.30\% \pm 0.07\%$ | $0.048 \pm 0.001$ | 36470 |
| Discrete S | $90.07\% \pm 0.32\%$ | $90.25\% \pm 0.06\%$ | $0.050 \pm 0.002$ | 13918 |
| Triangle $\eta$ | $90.03\% \pm 0.10\%$ | $90.04\% \pm 0.23\%$ | $0.068 \pm 0.002$ | 35310 |
| Baseline | $87.70\% \pm 0.56\%$ | $88.36\% \pm 0.13\%$ | $0.033 \pm 0.001$ | 57838 |

Table 1: Comparison between different cyclical training schedules (cycle length and learning rate are optimized using a grid search). Discrete schedules perform similarly, or slightly better than triangular. Additionally, discrete $S$ schedule leads to much wider minima for similar loss. Hessian norm is approximated on 1 out of 6 repetitions and measured at minimum value (usually endpoint of training).

lines (cyclic schedules) have the same ratio of learning rate to batch-size as each other throughout, and that their dynamics are very similar to each other. The same holds for the green and red lines, of constant batch size and learning rate. This supports the theoretical result of this paper, that the ratio $S/\eta$ governs the stationary distribution of SGD.

We note that it is not just in the stationary distribution that the exchangeability holds in these plots - it appears throughout training, highlighted especially in the cyclic schedules. We postulate that due to the scaling relation in the Fokker-Planck equation, the exchangeability holds throughout learning as well as just at the end, as long as the learning rate does not get so high as to ruin the approximations under which the Fokker-Planck equation holds. We note similar behaviour occurs also for standard learning rate annealing schedules, which we omit here for brevity.

### F.3   DETAILS ON MEMORIZATION EXPERIMENT

We report learning curves from memorization experiment with $0.0$ momentum, see Fig. 14. We additionally confirm similar results to the previous experiments and show correlation between batch size and learning rate ratio and norm of Hessian, see Fig. 15 without momentum (left) and with momentum $0.9$ (right).

### F.4   CYCLICAL BATCH AND LEARNING RATE SCHEDULES

It has been observed that a cyclic learning rate (CLR) schedule leads to better generalization (Smith, 2015). In Sec. 4.2 we demonstrated that one can exchange cyclic learning rate schedule (CLR) with cyclic batch size (CBS) and approximately preserve the practical benefit of CLR. This exchangeability shows that the generalization benefit of CLR must come from the varying noise level, rather than just from cycling the learning rate. To explore why this helps generalization, we run VGG-11 on CIFAR10 using 4 training schedules: we compared two discrete schedules (where either $\eta$ or $S$ switches discretely from one value to another between epochs) and two baseline schedules, one constant ($\eta$ is constant) and one triangle ($\eta$ is interpolated linearly between its maximum and minimum value). We track the norm of the Hessian and the training loss throughout training. Each experiment is repeated six times. For each schedule we optimize $\eta$ in $[1e-3, 5e-2]$ and cycle length in $\{5, 10, 15\}$ on a validation set. In all cyclical schedules the maximum value (of $\eta$ or S) is $5\times$ larger than the minimum value.

First, we observe that cyclical schemes oscillate between sharp and wide regions of the parameter space, see Fig. 16. Next, we empirically demonstrate that a discrete schedule varying either $S$ or $\eta$ performs similarly, or slightly better than triangular CLR schedule, see Tab. 1. Finally, we observe that cyclical schemes reach wider minima at the same loss value, see Fig. 16 and Tab. 1. All of the above suggest that by changing noise levels cyclical schemes reach different endpoints than constant learning rate schemes with same final noise level. We leave the exploration of the implications and a more thorough comparison with other learning schedules for future work.

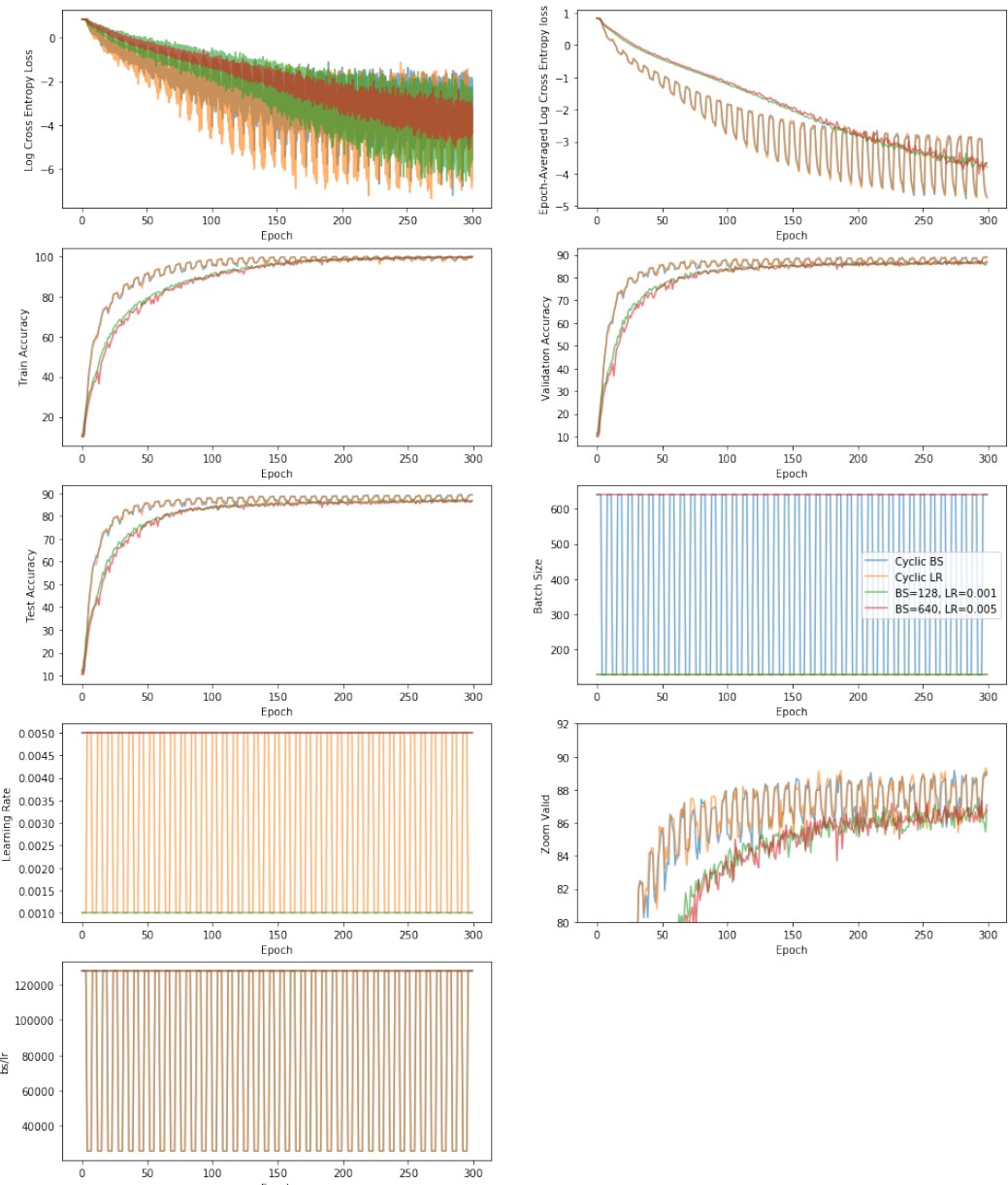

Figure 13: We show in more detail exchangeability of batch size and learning rate in a one-to-one ratio. In blue, cyclic batch size schedule between size 128 and 640 and fixed learning rate 0.005, is exchangeable with orange cyclic learning rate schedule between learning rates 0.001 and 0.005 with fixed batch size 128. In green, constant batch size 640 and constant learning rate 0.005 is exchangeable with, in red, constant batch size 128 and constant learning rate 0.005.

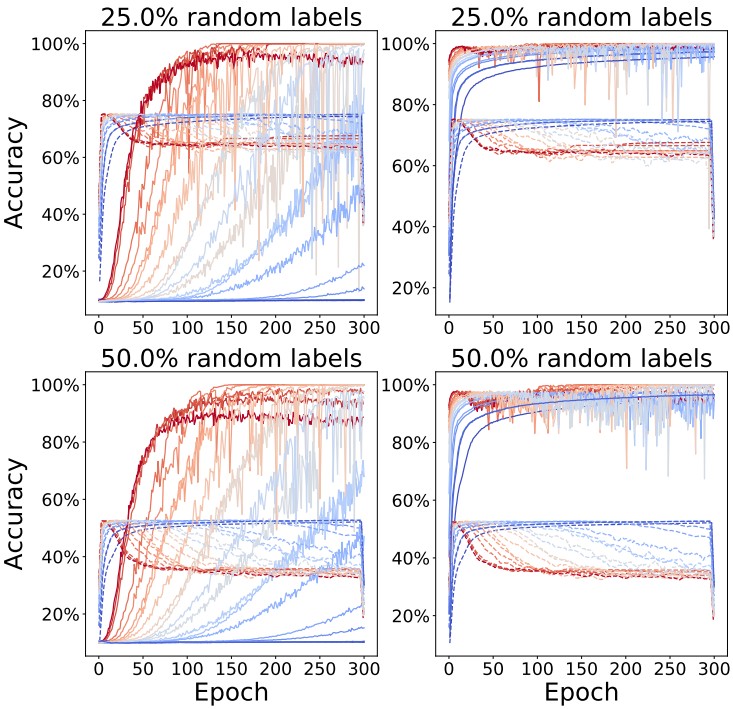

Figure 14: Learning curves for memorization experiment with momentum $0.0$. Solid lines represent training accuracy, dotted validation accuracy. Warm color indicates higher $\frac{\eta}{S}$ ratio.

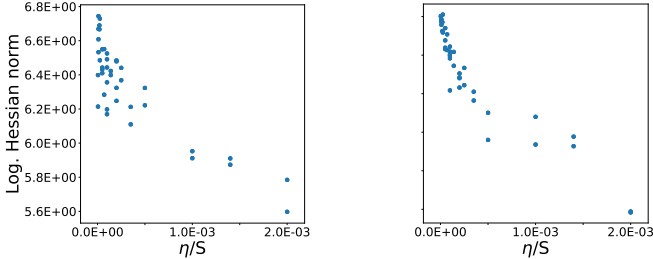

Figure 15: Correlation between (approximate) norm of Hessian of best validation minima and learning rate to batch-size ratio for $0.0$ (left) and $0.9$ (right) momentum.

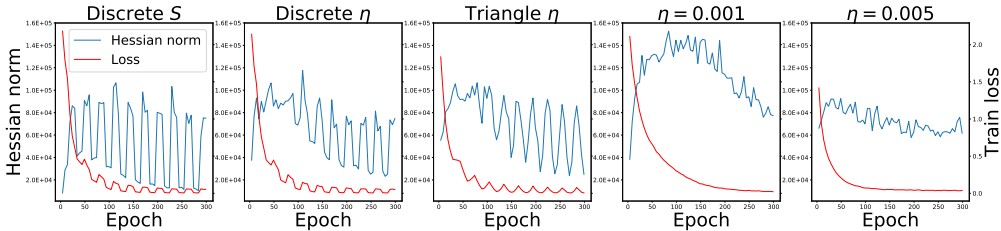

Figure 16: Cyclical schemes oscillate between sharp and wide regions. Additionally, cyclical schemes find wider minima than baseline run for same level of loss, which might explain their better generalization. All cyclical schedules use base $\eta = 0.005$ and cycle length 15 epochs, which approximate convergence at the end of each cycle. Plots from left to right: discrete $S$, discrete $\eta$, triangle $\eta$, constant learning rate $\eta = 0.001$, constant learning rate $\eta = 0.005$. On vertical axis we report loss (red) and approximated norm of Hessian (blue).

