# OpenReview forum: "Three factors influencing minima in SGD"
_ICLR.cc/2018/Conference — Reject_

### Official Review · AnonReviewer2 · 2017-11-26
**A stability analysis of local optima in constant-rate SGD**

**Rating:** 6
**Confidence:** 4

**Review:**

The paper investigates how the learning rate and mini-batch size in SGD impacts the optima that the SGD algorithm finds.
Empirically, the authors argue that it was observed that larger learning rates converge to minima which are more wide,
and that smaller learning rates more often lead to convergence to minima which are narrower, i.e. where the Hessian has large Eigenvalues. In this paper, the authors derive an analytical theory that aims at explaining this phenomenon.

Point of departure is an analytical theory proposed by Mandt et al., where SGD is analyzed in a continuous-time stochastic
formalism. In more detail, a stochastic differential equation is derived which mimicks the behavior of SGD. The advantage of
this theory is that under specific assumptions, analytic stationary distributions can be derived. While Mandt et al. focused
on the vicinity of a local optima, the authors of the present paper assumed white diagonal gradient noise, which allows to
derive an analytic, *global* stationary distribution (this is similar as in Langevin dynamics).

Then, the authors focus again on individual local optima and "integrate out" the stationary distribution around a local optimum, using again a Gaussian assumption. As a result, the authors obtain un-normalized probabilities of getting trapped in a given local optimum. This un-normalized probability depends on the strength of the value of the loss function in the vicinity of the optimum, the gradient noise, and the width of the optima. In the end, these un-normalized probabilities are taken as
probabilities that the SGD algorithm will be trapped around the given optimum in finite time.


Overall assessment:
I find the analytical results of the paper very original and interesting. The experimental part has some weaknesses. The paper could be drastically improved when focusing on the experimental part.

Detailed comments:

Regarding the analytical part, I think this is all very nice and original. However, I have some comments/requests:

1. Since the authors focus around Gaussian regions around the local minima, perhaps the diagonal white noise assumption could be weakened. This is again the multivariate Ornstein-Uhlenbeck setup examined in Mandt et al., and probably possesses an analytical solution for the un-normalized probabilities (even if the noise is multivariate Gaussian). Would the authors to consider generalizing the proof for the camera-ready version perhaps?

2. It would be nice to sketch the proof of theorem 2 in the main paper, rather than to just refer to the appendix. In my opinion, the theorem results from a beautiful and instructive calculation that should provide the reader with some intuition.

3. Would the authors comment on the underlying theoretical assumptions a bit more? In particular, the stationary distribution predicted by the Ornstein-Uhlenbeck formalism is never reached in practice. When using SGD in practice, one is in the initial mode-seeking phase. So, why is it a reasonable assumption to still use results obtained from the stationary (equilibrated) distribution which is never reached?


Regarding the experiments: here I see a few problems. First, the writing style drops in quality. Second, figures 2 and 3 are cryptic. Why do the authors focus on two manually selected optima? In which sense is this statistically significant? How often were the experiments repeated? The figures are furthermore hard to read. I would recommend overhauling the entire experiments section.

Details:

- Typo in Figure 2: ”with different with different”.
- “the endpoint of SGD with a learning rate schedule η → η/a, for some a > 0, and a constant batch size S, should be the same
  as the endpoint of SGD with a constant learning rate and a batch size schedule S → aS.” This is clearly wrong as there are many local minima, and running teh algorithm twice results in different local optima.  Maybe add something that this only true on average, like “the characteristics of these minima ... should be the same”.

---

> ### Author Response · Authors · 2017-12-15
> **Response to AnonReviewer2**
>
> We thank the reviewer for their interesting comments and observations and for their enthusiasm for our results.
>
> Analytical Part
>
> Response to point 1, whether we can generalize to non-diagonal white noise.
>
> In short, we believe generalization beyond the isotropic case is nontrivial, and we leave for future work. To clarify, in Mandt et al. they assume globally the Ornstein-Uhlenbeck (quadratic loss) setup (i.e they only consider one minimum), whereas for Theorem 2 we assume a series of minima, and then approximate the integral using the second order Taylor series locally near each minima, but not globally. (For Theorem 1 there is no restriction on the loss at all). The proof of Theorem 1 only strictly holds if the gradient noise is isotropic - in the non-isotropic case, the Fokker Planck equation will be a complicated partial differential equation which doesn’t have a closed form analytic stationary solution in general. Instead one would need numerical solutions of the PDE or further simplifying assumptions for an analytic solution. The solution may also depend on the path in parameter space through which the process evolves, unless further assumptions are made.
>
> Response to point 2, whether the proof of theorem 2 can appear in the main paper.
>
> We have decided to keep the proof of theorem 2 in the appendix, in response to AnonReviewer3 who suggested this proof is fairly standard, and also to keep the paper easy to read on a first pass without too much mathematical detail.
>
> Response to point 3, that the equilibrium distribution of the SDE is not reached in practice.
>
> We agree that the stationary distribution is not precisely arrived at in practice, but it can be approached to a good approximation if enough epochs have passed. On the other hand, we are not necessarily interested in exactly reaching the equilibrium distribution, we are more interested in sampling from the equilibrium distribution, which can happen in a fewer number of epochs than it takes for the probability distribution to approach it.
>
> Experimental Part
>
> In figures 2 and 3 we show a qualitative result, common in the literature, e.g. Fig. 3 of https://arxiv.org/pdf/1609.04836.pdf, which expresses intuitively the consistency of the theory with experiment. They are just a one-dimensional slice through parameter space and so should be treated with a pinch of salt. In the original submission there were five plots each of which shows the consistency of our experiments with our theory pictorially. To show we are not manually selecting minima that fit our claims, we have run some more interpolation experiments to validate this. In the new version we have added more seeds to show the robustness with respect to the model random initialization of the result, see Appendix F in the revised version.
>
> We have improved the quality of the figures to make them easier to read.
>
> On detail 1, the typo, we have fixed this in the new version.
>
> On detail 2, that the endpoints will not be the same, we agree with the reviewer here and thank them for the suggestion to clarify the phrasing in this way and have edited accordingly to read instead that the characteristics of these minima should be the same, not the actual minima. This is similar to the fix done for AnonReviewer3 on the vagueness of the phrase “the probability of ending in a certain minimum”.

---

### Official Review · AnonReviewer3 · 2017-11-27
**Interesting paper, however not convincing theoretical results**

**Rating:** 3
**Confidence:** 4

**Review:**

In this paper, the authors present an analysis of SGD within an SDE framework. The ideas and the presented results are interesting and are clearly of interest to the deep learning community. The paper is well-written overall.

However, the paper has important problems.

1) The analysis is widely based on the recent paper by Mandt et al. While being an interesting work on its own, the assumptions made in that paper are very strict and not very realistic. For instance, the assumption that the stochastic gradient noise being Gaussian is very restrictive and trying to justify it just by the usual CLT is not convincing especially when the parameter space is extremely large, the setting that is considered in the paper.

2) There is a mistake in the proof Theorem 1. Even with the assumption that the gradient of sigma is bounded, eq 20 cannot be justified and the equality can only be "approximately equal to". The result will only hold if sigma does not depend on theta. However, letting sigma depend on theta is the only difference from Mandt et al. On the other hand, with constant sigma the result is very trivial and can be found in any text book on SDEs (showing the Gibbs distribution). Therefore, presenting it as a new result is misleading.

3) Even if the sigma is taken constant and theorem 1 is corrected, I don't think theorem 2 is conclusive. Theorem 2 basically assumes that the distribution is locally a proper Gaussian (it is stated as locally convex, however it is taken as quadratic) and the result just boils down to computing some probability under a Gaussian distribution, which is still quite trivial. Apart from this assumption not being very realistic, the result does not justify the claims on "the probability of ending in a certain minimum" -- which is on the other hand a vague statement. First of all "ending in" a certain area depends on many different factors, such as the structure of the distribution, the initial point, the distance between the modes etc. Also it is not very surprising that the inverse image of a wider Gaussian density is larger than of a pointy one. This again does not justify the claims. For instance consider a GMM with two components, where the means of the individual components are close to each other, but one component having a very large variance and a smaller weight, and the other one having a lower variance and higher weight. With authors' claim, the algorithm should spend more time on the wider one, however it is evident that this will not be the case.

4) There is a conceptual mistake that the authors assume that SGD will attain the exact stationary distribution even when the SDE is simulated by the fixed step-size Euler integrator. As soon as one uses eta>0 the algorithm will never attain the stationary distribution of the continuous-time process, but will attain a stationary distribution that is close to the ideal one (of course with several smoothness, growth assumptions). The error between the ideal distribution and the empirical distribution will be usually O(eta) depending on the assumption and therefore changing eta will result in a different distribution than the ideal one. With this in mind the stationary distributions for (eta/S) and (2eta/2S) will be clearly different.


The experiments are very interesting and I do not underestimate their value. However, the current analysis unfortunately does not properly explain the rather strong claims of the authors, which is supposed to be the main contribution of this paper.

---

> ### Author Response · Authors · 2017-12-15
> **Response to AnonReviewer3 (Part I of III)**
>
> We thank the reviewer for their interest in our paper, and the detailed review. We will address each points of the review in turn, and supplement responses with experiments where possible. Before that, we wanted to stress here two crucial points about our submission:
>
> We would like to restate our main claim. This is that learning rate over batch size, along with noise in gradients, controls the stationary distribution from which SGD “samples” a solution. This claim (especially the importance of the ratio of learning rate to batch size) has not been made before. We discuss in the theory section, and in the experiment section how these findings are reflected in practice. Please refer to the rebuttal of AnonReviewer1 for more details about this point.
>
> Second, we believe our paper is  different from Mandt et al.. Our goal is comparing the different relative probabilities of ending in different “minima regions”, characterized by a loss value and hessian determinant. In particular we differ in Assumption 4 of Mandt et al. where in their whole analysis they restrict attention only to a region within a quadratic bowl, whereas we allow for a general loss function with multiple minima regions. In contrast, the goal of Mandt et al. is to show that under certain assumptions, SGD can be seen as sampling from a quadratic posterior (see for instance Fig.4 in Mandt et al), whereas we view SGD as sampling a solution from a stationary distribution that is not just a quadratic. For theorem 2, which talks about the probability of ending in a minima with certain characteristics, we use a Laplace approximation to evaluate an integral, which uses the second order Taylor expansion of the loss locally around a given minima - but this is not the same as the assumption in Mandt et al. which is that the loss is globally approximated by a quadratic. We have made changes to the paper at the end of Section 3.1 and in Appendix D to emphasize this.
>
> We have clarified the aforementioned points in the revised paper.
>
> Detailed responses:
> 1. Response to point 1, that assuming the batch gradient converges via the CLT to a Gaussian distribution.
>
> It is a common assumption that the stochastic gradient noise can be modelled as Gaussian, for instance in the paper by Li et al. ‘15, https://arxiv.org/pdf/1511.06251.pdf the stochastic differential equation that we use has been proven to approximate SGD in the weak sense. More precisely, the use of the central limit theorem is appropriate in this case: the minibatch samples are randomized draws from a fixed distribution with finite variance: the distribution over the randomly ordered full dataset. Typical minibatch sizes are large by any CLT standard.  The data exchangeability ensure that there is a shared variance, C, for all data points, and hence by the CLT the average over the minibatch will have variance C/S for a batch size S. We have produced a plot for the MLP model on the FMNIST dataset used in section 4.1 of the submitted paper which shows samples of gradients in randomly chosen directions for different batch sizes, which appear to follow a Gaussian distribution already for a typical batch size of 64. Here is a sample from 10 random directions at initialization https://anonfile.com/52v9m2d0b6/grid.pdf, and at best validation point https://anonfile.com/6cvcmbd4b8/grid_after_training.pdf.

---

> ### Author Response · Authors · 2017-12-15
> **Response to AnonReviewer3 (Part II of III)**
>
> 2. Response to paragraph 2, first point, that there is a mistake in Theorem 1:
>
> We agree there is a mathematical mistake in allowing sigma to vary with theta. We address this by changing our assumption so that sigma is constant. This modification does not affect our end results as the equilibrium distribution will then be the standard Gibbs distribution. Though the Gibbs distribution appears in the SDE literature, this exact expression has not appeared before in the machine learning literature to the best of our knowledge, explicitly showing  the dependence of the loss, learning rate, batch size and sigma.
>
> Response to paragraph 2, second point, that letting sigma depend on theta is the only difference to Mandt et al.:
>
> We agree with the reviewer that taking a constant sigma is the same as Assumption 2 of Mandt et al. However, this is not the only difference between our paper and Mandt et al.
> As stated on the first point of our introduction to this rebuttal, we do not assume Mandt et al. Assumption 4, which is the assumption that the iterates throughout are constrained to be in a region in which the loss surface is quadratic. Instead we allow iterates to be drawn from any region of parameter space for a general loss function. For Theorem 2 we decompose the whole loss surface into different basins of attraction of different sizes. For each of these different basins of attraction we use a second order Taylor expansion to evaluate the integral for the result of Theorem 2, allowing us to define the sizes of these basins and to compare between these basins. There is no comparison of different basins in Mandt et al. and this comparison is critical for the important observations about which minima SGD ends up in. To summarize, letting sigma be constant is mathematically necessary, but this was not the only difference between us and Mandt et al., instead our key difference is that we don’t restrict to a single basin with a quadratic loss, instead we consider many basins.
>
> 3. Response to paragraph 3, even if Theorem 1 is corrected, the reviewer thinks Theorem 2 is not conclusive. Let us address each point in turn:
>
> About the concern that Theorem 2 is quite trivial.
> We disagree that the result is trivial - it is indeed a simple calculation, but it is not obvious a priori that it will be the determinant of the Hessian that will appear in the prefactor, nor that the ratio of learning rate to batch size will control the weight given to width (from the Hessian prefactor) over depth.
>
> About "the probability of ending in a certain minimum" is vague.
> We agree that the concept of the size of minima SGD finishes in is indeed vague unless it is sufficiently well defined. To discuss this we propose an approximation of each minima region by the quadratic bowl at that minima, and the size of the minima by the effective posterior mass of the corresponding Gaussian distribution. This Laplace approximate mass has been used before in the context of computing the evidence in Bayesian methods: it is indeed approximate, but it is sufficiently well defined, and captures enough for us to be able to discuss the critical issue of sizes of minima regions. For example to calculate Bayes’ factors e.g. in Kass and Raferty https://www.stat.washington.edu/raftery/Research/PDF/kass1995.pdf or for Bayesian model comparison in Mackay https://pdfs.semanticscholar.org/e5c6/a695a4455a526ec8955dcc0fa2d6810089e9.pdf. We have revised the phrase to read instead “the probability of ending in a minimum characterized by a certain loss value and Hessian determinant”.
>
> With regards to the dependence of the endpoint on the initial point we point out that the stationary distribution theoretically doesn’t depend on the initialization, and this will be approximately true in practice if the algorithm is run for long enough.
>
> We would like to clarify the dependence on the distance between the modes. We point out in appendix D that we assume the modes are separated by a large enough distance that the tails of the Gaussian approximation do not contribute significantly. This means that the example the reviewer gives of a GMM with close means is not valid for our situation since we assume the modes are separated enough in the derivation in appendix D. We would be happy to promote details of this assumption to the main text for clarity.
>
> To summarize, our claim is not that the algorithm will spend more time on the wider minima, instead our claim is that the ratio of learning rate to batch size controls the tradeoff between width and depth, so whether it spends more time at a wider minima depends on the value of this ratio. Finally, we would like to emphasize that our empirical results confirm that a higher ratio of learning rate to batch size leads to a wider region being sampled.
>
> (continued...)

---

> ### Author Response · Authors · 2017-12-15
> **Response to AnonReviewer3 (Part III of III)**
>
> 4. In response to point 4, on the error between SGD and the SDE stationary solution. It is standard to approximate SGD with an SDE in the machine learning literature and use the stationary distribution as an approximation of the learnt distribution. We are aware that SGD will not exactly attain the SDE stationary distribution. Instead, we recognise the breakdown of the theory, and have a whole section devoted to it: In the new version this is Section 4.5 “Breaking of the theory in practice”, where we see this error for larger eta, as the reviewer states. We highlight other limitations in Appendix F. We also specifically mention the approximation holds only to first order in eta in the final paragraph of Appendix B. In the new version we give a more detailed experiment of the breakdown of the theory in figure 7. We hope this addresses the concern that there is a conceptual mistake - we are aware that SGD will not attain the exact stationary distribution for eta>0 and this is reflected in our paper.

---

### Official Review · AnonReviewer1 · 2017-11-27
**Theory not particularly novel, experiments okay.**

**Rating:** 5
**Confidence:** 4

**Review:**

The authors study SGD as a stochastic differential equation and use the Fokker planck equation from statistical physics to derive the stationary distribution under standard assumptions. Under a (somewhat strong) local convexity assumption, they derive the probability of arriving at a local minimum, in terms of the batchsize, learning rate and determinant of the hessian.

The theory in section 3 is described clearly, although it is largely known. The use of the Fokker Planck equation for stationary distributions of stochastic SDEs has seen wide use in the machine learning literature over the last few years, and this paper does not add any novel insights to that. For example, the proof of Theorem 1 in Appendix C is boilerplate. Also, though it may be relatively new to the deep learning/ML community, I don't see the need to derive the F-P equation in Appendix A.

Theorem 2 uses a fairly strong locally convex assumption, and uses a straightforward taylor expansion at a local minimum. It should be noted that the proof in Appendix D assumes that the covariance of the noise is constant in some interval around the minimum; I think this is again a strong assumption and should be included in the statement of Theorem 2.

There are some detailed experiments showing the effect of the learning rate and batchsize on the noise and therefore performance of SGD, but the only real insight that the authors provide is that the ratio of learning rate to batchsize controls the noise, as opposed to the that of l.r. to sqrt(batchsize). I wish this were analyzed in more detail.

Overall I think the paper is borderline; the lack of real novelty makes it marginally below threshold in my view.

---

> ### Author Response · Authors · 2017-12-15
> **Response to AnonReviewer1**
>
> We thank the reviewer for the insightful comments and interesting questions they pose. We think the paper will be stronger through the clarifications that ensue.
>
> 1)
>
> Response to the comments about the Fokker-Planck equation and the novelty of our results.
> We would like to clarify where our novelty arises. In our paper the main new result is that the learning rate over batch size ratio controls the tradeoff between the width (i.e. sharpness) and depth of the region in which SGD ends. In Theorem 1 and the surrounding text, we clarify the relationship between the batch size and learning rate and the effect that this has on the resulting equilibrium distribution. Though theorem 1 is a standard Gibbs-Boltzmann result, it is valuable to express it as a function of batch size and learning rate: this has not previously been emphasised in the literature, and it is this relationship that provides the insight into how SGD performs. To the best of our knowledge we have not seen the exact statement of Theorem 1 in the literature before. We do agree that a Gibbs distribution and its derivation has appeared before in the machine learning literature. For example, in the online setting we are aware of the results of equation (24) of Heskes and Kappen (http://www.sciencedirect.com/science/article/pii/S0924650908700382), but the relation here does not give the temperature of the Gibbs distribution in terms of the learning rate, batch size and gradient covariance. So we believe the result is novel in the machine learning context for minibatch stochastic gradient descent. We have adjusted the presentation of Theorem 1 to reflect this. We have also renamed the section from ‘Theoretical Results’ to ‘Insights from Fokker-Planck’ to reflect more clearly that our novelty is the insights gained rather than the derivation of new mathematical results.
>
> 2)
>
> Response to the comment on Theorem 2 that we assume the gradient covariance is constant in some region. We agree that the assumption should be included in the statement of Theorem 2, and further would like to revise Theorem 2 to be such that the covariance of the noise is constant and proportional to the identity everywhere. This stronger assumption corrects a mathematical mistake pointed out by AnonReviewer3 - nonetheless, this stronger assumption is sufficient for our requirements, to obtain an analytic solution for the stationary distribution. In the revised version this assumption appears in the statement of theorem 2.
>
> 3)
>
> Response to the comment “the only real insight that the authors provide is that the ratio of learning rate to batchsize controls the noise, as opposed to the that of l.r. to sqrt(batchsize)”.
>
> We disagree that the ratio of learning rate to batch size controlling noise being the only real insight. We agree this is a core contribution of our work. However, more than interpreting this ratio as just the noise, we also investigate how this ratio affects the geometry sampled by SGD, the learning dynamics, the memorization and generalization.
>
> We verify in the paper that when keeping the ratio of learning rate to batch size the same, we terminate in a region with similar properties, hessian, loss and performance. We did not focus on other scaling strategies such as square root as they did not appear in our theoretical analysis and investigations of them have appeared previously, e.g. in Hoffer et al., as referenced in Section 4.2. We would be happy to include further experiments on square root scaling if the reviewer suggests.

---

### Public Comment · ~Kaylee_Kutschera1 · 2017-12-14
**Reproducibility Report**


*Introduction*
We sought to reproduce the results of this paper. The three sections investigated are composed of sections 4.1, 4.3 and 4.5. The results of studying the effect of controllable noise, the effects of randomized labels, and the effect of cyclic learning rate on generalization were investigated.

*Reproducibility Methodology and Results*
Controllable Noise
For the first experiment, a 20 layer multilayer perceptron (MLP) with ReLU activation functions trained on the FashionMNIST dataset was used. The structure of the MLP is as described in the paper "Adding Gradient Noise Improves Learning for Very Deep Networks". The number of epochs used was not specified, so 15 epochs was used in our network since it is the number of epochs used in other sections of the paper. Since the results were meant to hold for all datasets, the MLP was also tested on the MNIST dataset. For both datasets, the network was trained on all data, excluding the 10,000 test images. The hypothesis regarding the correlation between the controllable noise and test accuracy appeared to hold on both datasets.
The second reproduction used VGG-11 architecture on the CIFAR-10 dataset. Since no other information was provided about the network, the paper "Very Deep Convolutional Networks for Large-Scale Image Recognition" was used to build VGG-11. Due to the multiple max-pool layers, the architecture was not compatible with CIFAR-10. As a result this reproduction was dropped.

Memorization
A 2 layer multilayer perceptron was used to test the memorization phenomenon. The network ran on 300 epochs with ReLU activation functions, 256 units per layer, no momentum, and using the digit MNIST dataset. The network did not learn the data when using the original learning rates, so smaller learning rates were used. To save time, two extreme ratios using the batch sizes (50, 800) and learning rates (0.005, 0.01) were utilized.
A varying number of epochs were required to learn different sized subsets of the data. In the case of the entire dataset, 1000 epochs were still not enough to learn the data. For a subset of 1000 training and validation data points, it was found that higher ratio of learning rate to batch size generalizes better. However, with the addition of more data points, the results achieved worsened.
Networks with a smaller ratio learned more quickly than those with a larger ratio. When testing on 7000 data points (using a learning rate of 0.05 with 700 and 10 epochs) the smaller ratio finished training, while the greater ratio did not. This reinforces the idea that a smaller ratio memorizes more.

Cyclic Learning Rate
In section 4.5 of the original paper, the effect of CLR on testing accuracy was experimented on the CIFAR-10 dataset using the VGG-11 architecture. However, as mentioned before, they are not compatible. FashionMNIST was used as an alternative dataset and the previously mentioned 20 layer MLP with ReLU activation was used.
Some hyperparameters of CLR are missing such as higher bound and lower bound for learning rates. After experimentation, an optimum result was achieved with higher bound at 0.002 and lower bound at 0.001. To compare the effect of CLR on generalization, constant learning rates of 0.001, 0.002 and 0.0015 using a mini-batch size of 8 were tested as they represent a relatively similar structure to a CLR with step size of 4. From our results, CLR improves generalization as stated in the original paper and furthermore, CLR achieved the highest accuracy among all the accuracies that we obtained using constant learning rate.

*Conclusion*
We were able to reproduce the experiment regarding the controllable noise.
In the case of memorization, only in the cases with small subsets were the results vaguely be reproduced. While the results were not able to be recreated, in some cases the results acquired supported the authors claims about the ratio: the smaller ratio learned faster than the bigger ratio, suggesting that having a smaller ratio leads to faster memorization. But overall, it cannot be said that the results achieved in this paper were able to properly replicate those of the original. In the case of cyclic learning rate, although not being able to reproduce the exact experiment, the conclusions reached show that CLR does improve generalization on FashionMNIST when compared with constant learning rate.

To see full references and more details, view the full report by S. Huang, K. Kutschera, and S. Perry-Fagant : http://cs.mcgill.ca/~kkutsc/reproduce.pdf.

---

> ### Author Response · Authors · 2017-12-15
> **Thanks for Reproducing**
>
> First of all we would like to thank authors for reproducing our results, we are very happy to see interest in our work! We will do our best to further investigate the report soon. There are some issues that need to be clarified (e.g. x axis of memorization experiment is different than in our paper), we contacted authors of reproduction via e-mail to clarify.
>
> In the meantime, let us clarify cyclical batch size. In our submission we plot batch size and learning rate over time for both schedules in the Appendix.
> We also do mention in text it is just replacing any relative change in learning rate with batch size change, for instance if learning rate is increased by factor of 5 (e.g. 0.1 to 0.5), we replace it with reduction of batch size by factor of 5 (e.g. 100 to 25). Adding to your report results of CBS, especially discrete one, would be very interesting. We will clarify it further in text.

---

### Author Response · Authors · 2017-12-15
**Explanation of changes from original submission**

We added many clarifying changes, including discussions, better plots, and some improvements to experiments (e.g. larger grid in “Breaking point” section). This increased submission size by 2 pages, but we believe it was necessary to address all reviewer’s points. We would be grateful for feedback if some clarifications are too explicit, or if we should reduce the size of submission to previous size. Easiest way of reducing size would be moving some of the enlarged and expanded figures to Appendix.

Changes:
We revised abstract to reflect better our novelty and main contribution
We added paragraph in Related work on Fokker-Planck equation
We improved figures as suggested by reviewers, e.g. Figure 4 and Figure 6 are enlarged.
We renamed section to 3 from “Theoretical results” to “Insights from Fokker-Planck” and 3.2 from “Main results” to “Three factors influencing equilibrium distribution”, to reflect the novel main finding of this section.
We reworded significantly section 3, mostly in response to reviewer 2:
We added many clarifications, e.g. in opening of section 3 we say “We make the assumption of isotropic covariance (...)”, or we added whole paragraph discussing Theorem 1 in 3.2. At the end of section 3.1 we add a clarifying remark on how we differ to Mandt et al. at the end of section 3.1. and a reference to Li et al. justifying the approximation of SGD by an SDE.
Added discussion sections after each theorem, which talk about the assumptions and interpretations of the results.
We made changes to theory
We fixed assumption of Theorem 1 as suggested by reviewer 2 to have a constant sigma.
We clarified that we assume equilibrium distribution in solution of Fokker-Planck rather than just the stationary distribution.
In 4.1 we rerun MLP experiment on a 4-layer network with Batch Normalization that is closer to assumptions made in Theorem 1, and we moved the 20 layer network without Batch Normalization experiments to appendix. Correlations remain qualitatively similar between the two experiments.
In 4.2 (“eta/S determines learning dynamics of SGD”) we added clarifying paragraph discussing that theory predicts “invariance” of endpoint of SGD, while dynamics “invariance” is an additional experimental result
In 4.3 (“impact of SGD on memorization”) we added minor clarifications
In 4.4 (“Breaking point of the theory in practice”) we significantly improved experiment by running larger grid, and improving plots
We added Section 5 “Discussion” which in 4 paragraphs summarizes results.
Due to large space taken by all of the above changes we moved 4.5 (“Cyclical batch and learning rate schedule”) to Appendix, and referred to it in 4.2, which also discussed cyclical batch size and learning rate schedules.
We also included some changes in 4.5 (now in Appendix). We changed tracking ratio of hessian and loss, to tracking hessian. We rerun larger grid search and included table comparing performance of discussed schedules (CLR, CBS and constant).

---

### Decision · Program_Chairs · 2018-01-29
**ICLR 2018 Conference Acceptance Decision**

**Decision:**

Reject

**Comment:**

Dear authors,

The reviewers agreed that the theoretical part lacked novelty and that the paper should focus on its experimental part which at the moment is not strong enough to warrant publication.

Regarding the theoretical part, here are the main concerns:
- Even though it is used in previous works, the continuous time approximation of stochastic gradient overlooks its practical behaviour, especially since a good rule of thumb is to use as large as stepsize as possible (without reaching divergence), as for instance mentioned in The Marginal Value of Adaptive Gradient Methods in Machine Learning by Wilson et al.
- The isotropic approximation is very strong and I don't know settings where this would hold. Since it seems central to your statements, I wonder what can be deduced from the obtained results.
- I do not think the Gaussian assumption is unreasonable and I am fine with it. Though there are clearly cases where this will not be true, it will probably be OK most of the time.

I encourage the authors to focus on the experimental part in a resubmission.